# Tyrosine dephosphorylated cortactin downregulates contractility at the epithelial zonula adherens through SRGAP1

Xuan Liang[1], Srikanth Budnar[1], Shafali Gupta[1], Suzie Verma[1], Siew-Ping Han[1], Michelle M. Hill[2], Roger J. Daly[3], Robert G. Parton [1,4,5], Nicholas A. Hamilton[1], Guillermo A. Gomez[1,6] & Alpha S. Yap[1,4]

Contractile adherens junctions support cell–cell adhesion, epithelial integrity, and morphogenesis. Much effort has been devoted to understanding how contractility is established; however, less is known about whether contractility can be actively downregulated at junctions nor what function this might serve. We now identify such an inhibitory pathway that is mediated by the cytoskeletal scaffold, cortactin. Mutations of cortactin that prevent its tyrosine phosphorylation downregulate RhoA signaling and compromise the ability of epithelial cells to generate a contractile zonula adherens. This is mediated by the RhoA antagonist, SRGAP1. We further demonstrate that this mechanism is co-opted by hepatocyte growth factor to promote junctional relaxation and motility in epithelial collectives. Together, our findings identify a novel function of cortactin as a regulator of RhoA signaling that can be utilized by morphogenetic regulators for the active downregulation of junctional contractility.

[1] Division of Cell Biology and Molecular Medicine, The University of Queensland, St. Lucia, QLD 4072, Australia. [2] The University of Queensland Diamantina Institute, Brisbane, QLD 4102, Australia. [3] Cancer Program, Biomedicine Discovery Institute and Department of Biochemistry and Molecular Biology, Monash University, Clayton, VIC 3800, Australia. [4] Program in Membrane Interface Biology, Institute for Molecular Bioscience, The University of Queensland, St. Lucia, QLD 4072, Australia. [5] Centre for Microscopy and Microanalysis, The University of Queensland, St. Lucia, QLD 4072, Australia. [6]Present address: Centre for Cancer Biology, SA Pathology and the University of South Australia, Adelaide, SA 5000, Australia. Correspondence and requests for materials should be addressed to G.A.G. (email: guillermo.gomez@unisa.edu.au) or to A.S.Y. (email: a.yap@uq.edu.au)

E pithelial adherens junctions are contractile structures, where coupling of actomyosin to E-cadherin generates junctional tension that promote cell–cell adhesion and assembly of the specialized adherens junction of the zonula adherens (ZA)[1, 2]. In addition, the coupling of contractility to adhesion participates in a variety of morphogenetic processes, such as apical constriction and epithelial furrowing[3, 4].

The functional consequences of applying contractile force at junctions have commonly been studied when those forces are increased in some regulated fashion, or when coupling of contractility to adhesion is developmentally activated[3]. However, other developmental circumstances entail the downregulation of cell–cell junctions. In the extreme case, cell–cell contacts may break down altogether when E-cadherin expression is suppressed

**Fig. 1** Tyrosine non-phosphorylated cortactin disrupts zonula adherens (ZA) contractility. Cortactin (Cort) was depleted by shRNA (KD) and reconstituted with shRNA resistant cortactin variants (WT/3YF/3YD/W22A) fused to mCherry by lentiviral transduction. Transduced Caco-2 cells are indicated with *asterisks. Arrowheads* indicate homotypic junctions. **a** and **b** Effect on junctional tension. Recoil (**a**) and initial recoil measurements (**b**) of cell–cell junctions upon laser ablation. **c–f** Representative maximum projection view of the three apical-most sections (**c**) and quantification of apical junctional E-cadherin (E-cad; **d**), F-actin (**e**) and non-muscle myosin IIA (NMIIA; **f**). **g**, **h** FRAP **g** and immobile fractions (**h**) of E-cad-GFP at the ZA. $N = 3$ independent experiments. Data are means ± s.e.m.; one-way ANOVA with Dunnett's post hoc analysis; n.s., not significant; *$p < 0.05$; **$p < 0.01$; ***$p < 0.001$; ****$p < 0.0001$. *Scale bars* = 25 μm

during epithelial-to-mesenchymal transitions[5]. However, there are many other instances where cells rearrange while maintaining E-cadherin-based contacts with one another[4]. For example, when border cell clusters migrate in the *Drosophila* egg chamber[6], E-cadherin contacts persist between border cells and the nurse cells that they move through and are, indeed, necessary for invasive movement to occur[7]. Similarly, functional downregulation of adherens junctions is thought to underlie the morphogenetic changes seen when cultured mammalian epithelial cells are stimulated with Hepatocyte Growth Factor (HGF)[8, 9], which plays a vital role in organ development and wound repair[10, 11]. However, whether junctional contractility might also be modulated in these circumstances remains an open question.

In cultured epithelial cells, biogenesis of the junctional actomyosin cytoskeleton is necessary for the generation of contractility. This involves diverse processes that must be coordinated at the junctional cortex, including actin assembly[12, 13], filament network reorganization[14], and activation of non-muscle myosin II (NMII) by junctional RhoA[15]. Cortactin is a scaffolding protein that bears multiple potential protein−protein interaction domains and can influence many steps in cytoskeletal biogenesis[16]. It associates with the E-cadherin molecular complex and concentrates at sites of junctional con-tractility, notably when epithelia assemble a ZA, where it pro-motes actin assembly[17, 18]. Thus, cortactin presents as an attractive candidate to regulate actomyosin at the junctional cortex.

Cortactin is a tyrosine and serine phosphoprotein. Originally identified as a substrate for Src family kinases (SFK), cortactin is targeted by a number of protein kinases and phosphatases that function in different cellular processes[16]. Tyrosine phosphorylated cortactin is readily detected at cell−cell junctions, potentially generated by SFK activity in this location[19]. Indeed, expression of phosphomimetic mutants suggested that tyrosine phosphorylated cortactin might support junctional integrity downstream of junctional Src signaling[20, 21]. But how the tyrosine phosphorylated status of cortactin influences junctional biology remains poorly characterized. Here, we have identified a novel role for the tyrosine-dephosphorylated form of cortactin as a negative regulator of junctional contractility. We report that tyrosine-dephosphorylated cortactin downregulates junctional RhoA sig-naling by promoting the junctional accumulation of SRGAP1, a RhoA antagonist. We further show that this pathway is utilized by HGF to relax junctions and promote epithelial locomotility.

## Results

**Tyrosine non-phosphorylated cortactin downregulates ZA tension.** To begin, we tested how depleting cortactin affected junctional contractility in Caco-2 cells. Lentiviral shRNA reduced cellular cortactin (Supplementary Fig. 1a) and junctional cortactin staining detectable by immunofluorescence (IF; Supplementary Figs. 1d, e and 2) by ∼ 90%. We then used laser ablation to cut junctions marked by E-cad-GFP (expressed on an E-cad shRNA background; Fig. 1a) and measured the instanta-neous velocity of recoil as an index of tension (Fig. 1b)[15]. As previously reported[17, 18], cortactin knockdown (KD) decreased E-cadherin concentration at the apical ZA (Fig. 1c, d) without altering overall cellular or surface levels of the protein (Supple-mentary Fig. 1a, b). Fluorescence recovery after photobleaching (FRAP) revealed that the immobile fraction of E-cad-GFP (tagged at the endogenous locus by CRISPR-based genome editing; see Supplementary Methods) was also reduced by cortactin KD (Fig. 1g, h), suggesting that cortactin was required for E-cadherin stability at the ZA. Nonetheless, we were readily able to track the

recoil of junctional vertices after laser ablation. This revealed that initial recoil velocity was substantially reduced by cortactin KD (Fig. 1a, b, Supplementary Table 1). Junctional recoil, steady-state E-cadherin concentration and E-cad-GFP stability were restored to cortactin KD cells by expression of an RNAi-resistant wild-type (WT) cortactin transgene (Fig. 1a–d, g, h, Supplementary Fig. 1a, d, e), confirming that the effects were specific for change in cortactin.

Since recoil can be influenced by viscous drag (friction), as well as by changes in contractile tension[22, 23], we then modeled recoil as a Kelvin-Voigt fiber and extracted the rate constants ($k$-values) of relaxation (see Supplementary Methods). The $k$-values were not significantly altered in our experiments (Supplementary Table 1), arguing that the observed decrease in junctional recoil principally reflected a decrease in contractility in cortactin KD cells. This conclusion was further supported by the observation that junctional F-actin (Figs. 1c, e) and NMIIA (Figs. 1c, f) were also reduced by cortactin KD. Overall, these observations indicated that by some mechanism(s) cortactin contributed to junctional contractility.

Cortactin bears three tyrosine residues (Y421, 470, 486) that are substrates for phosphorylation. Of these, Y421 is most heavily phosphorylated and is necessary for other tyrosine residues to be phosphorylated[24]. As previously reported[19], pY421-cortactin (pY-cortactin) concentrates at the ZA (Supplementary Fig. 1c). To test whether the phosphorylation state of cortactin could affect junctional contractility, we expressed cortactin mutants in cortactin KD cells by lentiviral transduction (Supplementary Figs. 1a, d, e and 2). We focused, in particular, on 3YF cortactin, a mutant lacking the canonical tyrosine residues, and 3YD cortactin that is a putative phosphomimetic mutant. Both 3YF cortactin and 3YD cortactin localized to junctions as effectively as WT cortactin (Supplementary Fig. 1d, f). Furthermore, immunostain-ing revealed that all the transgenes restored cortactin staining at junctions to the same level as seen for endogenous cortactin in control-transfected cells (Supplementary Fig. 1d, e). Junctional staining for pY421 cortactin was also effectively abolished by cortactin KD and restored by WT cortactin (Supplementary Fig. 1g–j). However, although the transgenes were expressed at junctions to similar levels, pY421-staining was reduced in cells expressing 3YF cortactin compared with either WT or 3YD cortactin (some residual staining may reflect background staining, Supplementary Fig. 1g–j).

Strikingly, however, contractility was not restored when cortactin KD cells were reconstituted with 3YF cortactin (Fig. 1a, b). Junctional recoil (Fig. 1a, b) remained reduced in 3YF cortactin cells compared with WT cortactin (Fig. 1a–d) or with 3YD cortactin (Fig. 1a–d). As the $k$-values were not significantly altered (Supplementary Table 1), these findings implied that tyrosine non-phosphorylated cortactin compromised contractility by some mechanism. Consistent with this, junctional NMIIA levels remained reduced in cortactin KD cells expressing 3YF cortactin compared with either WT or 3YD cortactin (Fig. 1c, f, Supplementary Fig. 2c). Steady-state E-cadherin levels (Fig. 1d) and the immobile fraction of E-cad-GFP (Fig. 1g, h) at the ZA were also reduced in 3YF-reconstituted cells, in line with earlier evidence that NMII stabilizes E-cadherin to form the ZA[2, 25]. We also found that cell-matrix adhesion was reduced in cortactin KD cells; however, this was restored as effectively by 3YF cortactin as by WT or 3YD cortactin (Supplementary Fig. 3c). This suggested that tyrosine non-phosphorylated cortactin had a relatively selective effect on junctional contractility.

**RhoA signaling is downregulated at the ZA.** One possible mechanism to explain how the tyrosine phosphorylation status of

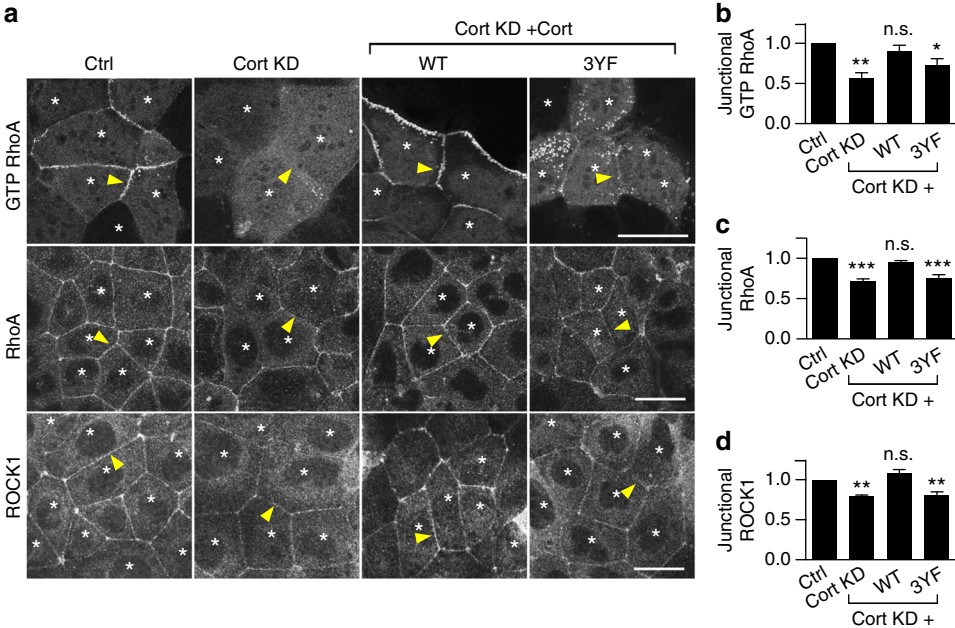

**Fig. 2** Tyrosine non-phosphorylated cortactin degrades junctional RhoA signaling. Representative images and quantification of fluorescence intensity of GTP RhoA (assessed by the ratio of mean fluorescence intensities of GFP-AHPH at the junctions/cytoplasm; **a**, **b**), RhoA (**a**, **c**), and ROCK1 (**a**, **d**) at apical junctions. Cortactin (Cort) was depleted by shRNA (KD) and reconstituted with shRNA resistant cortactin variants (WT/3YF) fused to mCherry by lentiviral transduction. Transduced cells are indicated with *asterisks. Arrowheads* indicate homotypic junctions. N = 3 independent experiments, data are means ± s.e.m.; one-way ANOVA with Dunnett's post hoc analysis; n.s., not significant; *$p < 0.05$; **$p < 0.01$; ***$p < 0.001$. *Scale bars* = 25 μm

cortactin might affect junctional contractility lay in the regulation of F-actin. Cortactin localizes the Arp2/3 complex to adherens junctions to support the actin assembly that is necessary for biogenesis of the actomyosin apparatus (Supplementary Fig. 3d, e)[12, 14]. Indeed, reconstitution of cortactin KD cells with a mutant (W22A) that cannot associate with Arp2/3[26, 27] failed to restore junctional tension (Supplementary Fig. 3a, b; Supplementary Table 1), steady-state E-cadherin levels (Fig. 1c, d, Supplementary Fig. 2a), or its stability (Fig. 1g, h). Consistent with its impact on actin assembly, junctional F-actin (Fig. 1c, e, Supplementary Fig. 2b) and NMIIA levels (Fig. 1c, f, Supplementary Fig. 2c) were not restored by W22A cortactin. This confirmed that actin assembly represents one pathway for cortactin to support junctional contractility. Similarly, W22A cortactin did not restore cell-substrate adhesion to cortactin KD cells (Supplementary Fig. 3c). Surprisingly, however, junctional F-actin (Fig. 1c, e, Supplementary Fig. 2b) and Arp3 (Supplementary Fig. 3d, e) were restored as effectively by 3YF cortactin as they were by WT or 3YD cortactin. Thus, the impaired contractility associated with 3YF cortactin was not readily explained by a defect in actin assembly.

We then considered whether tyrosine non-phosphorylated cortactin might affect junctional RhoA signaling, which is also essential for effective contractility at the ZA[15]. We tested this using GFP-AHPH, a location biosensor for endogenous active, GTP-RhoA that is derived from the C-terminus of anillin[28, 29]. In control cells, the ZA is one of the prominent sites for GTP-RhoA, as detected with GFP-AHPH (Fig. 2a)[29]. However, junctional GFP-AHPH was substantially reduced in cortactin KD cells (corrected for differences in cellular expression of GFP-AHPH) and this was restored by expression of WT cortactin (Fig. 2a, b). The effect of cortactin KD was confirmed using a Fluorescence resonance energy transfer (FRET)-based RhoA activity sensor[30] (Supplementary Fig. 3f, g). As a corollary, we measured the steady-state levels of junctional RhoA detectable after TCA fixation (Fig. 2a), of which ~50% required that RhoA be active,

being sensitive to C3-transferase (Supplementary Fig. 3h, i). Cortactin KD reduced junctional TCA-fixed RhoA (Fig. 2a, c) and also the RhoA effector, ROCK1 (Fig. 2a, d), whose junctional localization requires RhoA signaling[29]. Together, these assays suggested that cortactin was necessary to support the stable RhoA signaling zone of the ZA.

However, expression of 3YF cortactin did not restore junctional GTP-RhoA (Fig. 2a, b), despite being able to rescue F-actin (Fig. 1c, e). Similarly, both junctional RhoA (Fig. 2a, c) and ROCK1 levels (Fig. 2a, d) remained reduced in cortactin KD cells expressing 3YF cortactin, although their total protein levels were unaffected (Supplementary Fig. 3j). Together, these observations implied that the tyrosine phosphorylation status of cortactin could selectively influence RhoA signaling by a pathway that was independent of cortactin's ability to regulate steady-state F-actin at the junctions. We hypothesized that this disjunction might reflect a mechanism that allowed tyrosine non-phosphorylated cortactin to suppress RhoA signaling.

**Tyrosine non-phosphorylated cortactin recruits SRGAP1.** As cortactin does not bear any known RhoA-regulatory domains, we performed a mass spectrometry-based interaction screen comparing WT cortactin with 3YF cortactin to pursue how tyrosine non-phosphorylated cortactin might impair RhoA signaling. This identified SLIT-ROBO Rho GTPase activating protein 1 (SRGAP1) as a protein that interacted preferentially with 3YF cortactin rather than with WT cortactin (not shown). We corroborated this by expressing SRGAP1-GFP and mCherry-tagged cortactin mutants in HEK293 cells (Fig. 3a, b). This showed that SRGAP1-GFP bound 3YF cortactin to a greater extent than to either WT or 3YD cortactin (Fig. 3a, b).

To confirm this, we expressed cortactin-GFP in HEK293 cells and affinity-isolated it by GFP-Trap (Fig. 3c). The isolated cortactin-GFP was then dephosphorylated by incubation with λ–phosphatase (Fig. 3c, d). Conversely, to increase its tyrosine phosphorylation, cortactin-GFP was co-transfected with

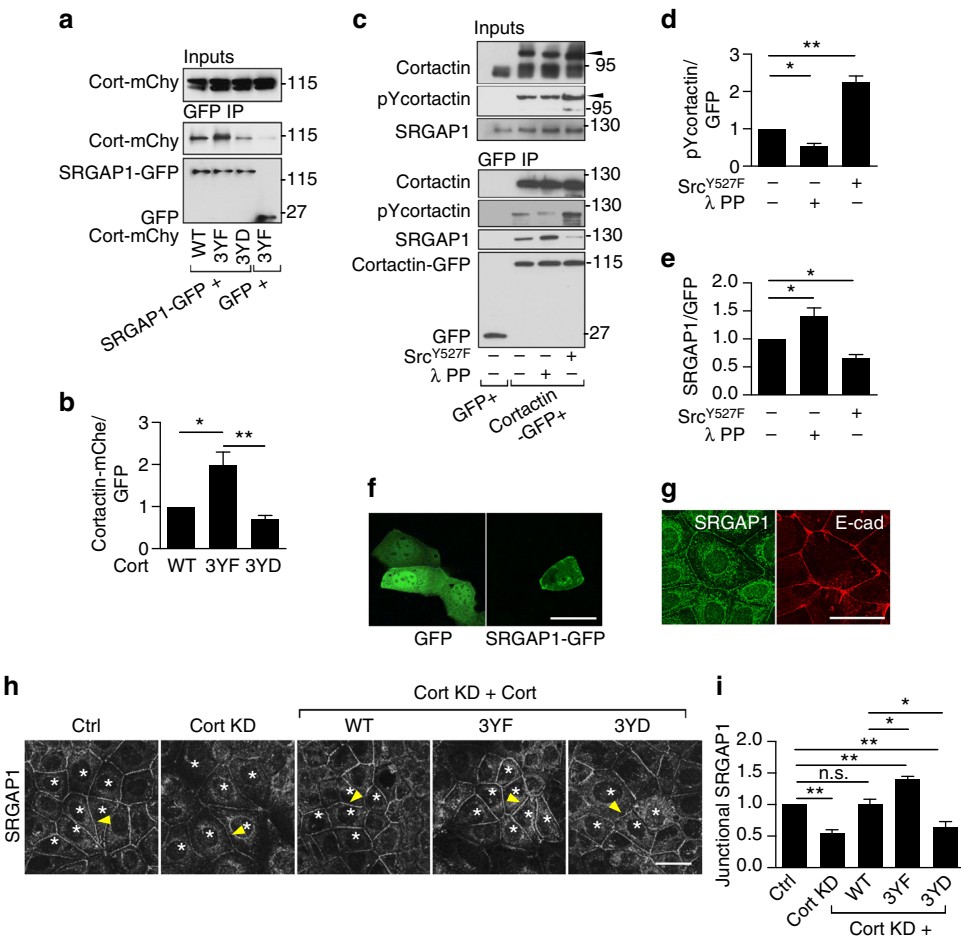

**Fig. 3** Tyrosine non-phosphorylated cortactin recruits SRGAP1 to junctions. **a–e** SRGAP1 preferentially interacts with tyrosine non-phosphorylated cortactin. **a** GFP-trap of HEK293T cells co-transfected with GFP or SRGAP1-GFP and WT/3YF/3YD cortactin-mCherry (cort-mChy). Transgenes were identified with anti-GFP or anti-mCherry antibodies, as appropriate. Molecular weight marker units, kDa. **b** Quantification of the relative cortactin binding to SRGAP1 normalized to SRGAP1-GFP expression levels. **c** GFP-trap of HEK293T cells co-transfected with GFP, or cortactin-GFP with or without Src$^{Y527F}$. The GFP-trap beads with cortactin-GFP were treated with lambda protein phosphatase (λ PP). The treated or untreated beads were incubated with cell lysates from Caco-2 cells previously transfected with cortactin siRNA. Transgenes were identified with the anti-GFP antibody. *Arrowheads* mark cortactin-GFP. Molecular weight markers unit, kDa. **d** Quantification of tyrosine 421-phosphorylated (pY) cortactin precipitated by GFP-trap. **e** Quantification of the relative SRGAP1 binding to cortactin normalized to cortactin-GFP expression levels. **f, g** SRGAP1 localizes at E-cadherin junctions in Caco-2 cells. **f** Live cell imaging of GFP and SRGAP1-GFP, and **g** dual-color immunofluorescence images of SRGAP1 and E-cadherin. **h** Representative images and **i** quantification of junctional fluorescence intensity of SRGAP1 in cortactin shRNA Caco-2 cells reconstituted with shRNA resistant WT/3YF/3YD cortactin (*asterisks*). *Arrowheads* indicate homotypic junctions. $N = 3$ independent experiments, data are means ± s.e.m.; one-way ANOVA with Dunnett's post hoc analysis; n.s., not significant; *$p < 0.05$; **$p < 0.01$; ****$p < 0.0001$. *Scale bars = 25 μm*

constitutively-active Src$^{Y527F}$ before affinity-isolation (Fig. 3c, d). Western analysis for pY421 confirmed the efficacy of these maneuvers (Fig. 3c, d). Isolated cortactin-GFP was then incubated with lysates from cortactin KD Caco-2 cells (to enhance the pool of SRGAP1 not bound to endogenous cortactin). The amount of SRGAP1 that precipitated with cortactin-GFP was reduced when cortactin-GFP was tyrosine phosphorylated with Src$^{Y527F}$ (Fig. 3c, e), whereas in vitro dephosphorylation significantly increased the amount of SRGAP1 that interacted with cortactin-GFP (Fig. 3c, e). Together, these findings indicate that SRGAP1 preferentially associates with tyrosine-dephosphorylated cortactin and this interaction is reduced when cortactin becomes phosphorylated.

SRGAP1 is a multi-domain protein that was first discovered as a component of the SLIT-ROBO pathway responsible for cell contact-dependent repulsion during neuronal guidance[31]. To better understand its potential role in mammalian epithelia, we then tested its subcellular localization. Live imaging revealed that SRGAP1-GFP, but not GFP alone, localized to apical cell–cell

contacts in Caco-2 cells (Fig. 3f, Supplementary Fig. 4a) and immunofluorescence staining for the endogenous protein showed SRGAP1 staining with E-cadherin at the ZA (Fig. 3g). Transient expression of mutant transgenes indicated that individual deletion of the F-BAR-FX, GAP or SH3 domains did not affect its junctional localization (Supplementary Fig. 4c–e), consistent with earlier evidence that multiple domains localize the *Caenorhabditis elegans* homolog, SRGP1, to adherens junctions[32]. In contrast, neither SRGAP1-GFP nor endogenous SRGAP1 were detectable at focal adhesions in confluent monolayers (Supplementary Fig. 4a, b).

To test whether tyrosine non-phosphorylated cortactin can influence the junctional levels of SRGAP1, we then examined SRGAP1 localization in cortactin KD cells reconstituted with phospho-mutant transgenes. Junctional SRGAP1 was reduced by cortactin KD and restored by WT cortactin (Fig. 3h, i). However, expression of 3YF cortactin increased junctional SRGAP1 above that of control or WT cortactin cells (Fig. 3h, i). Expression of

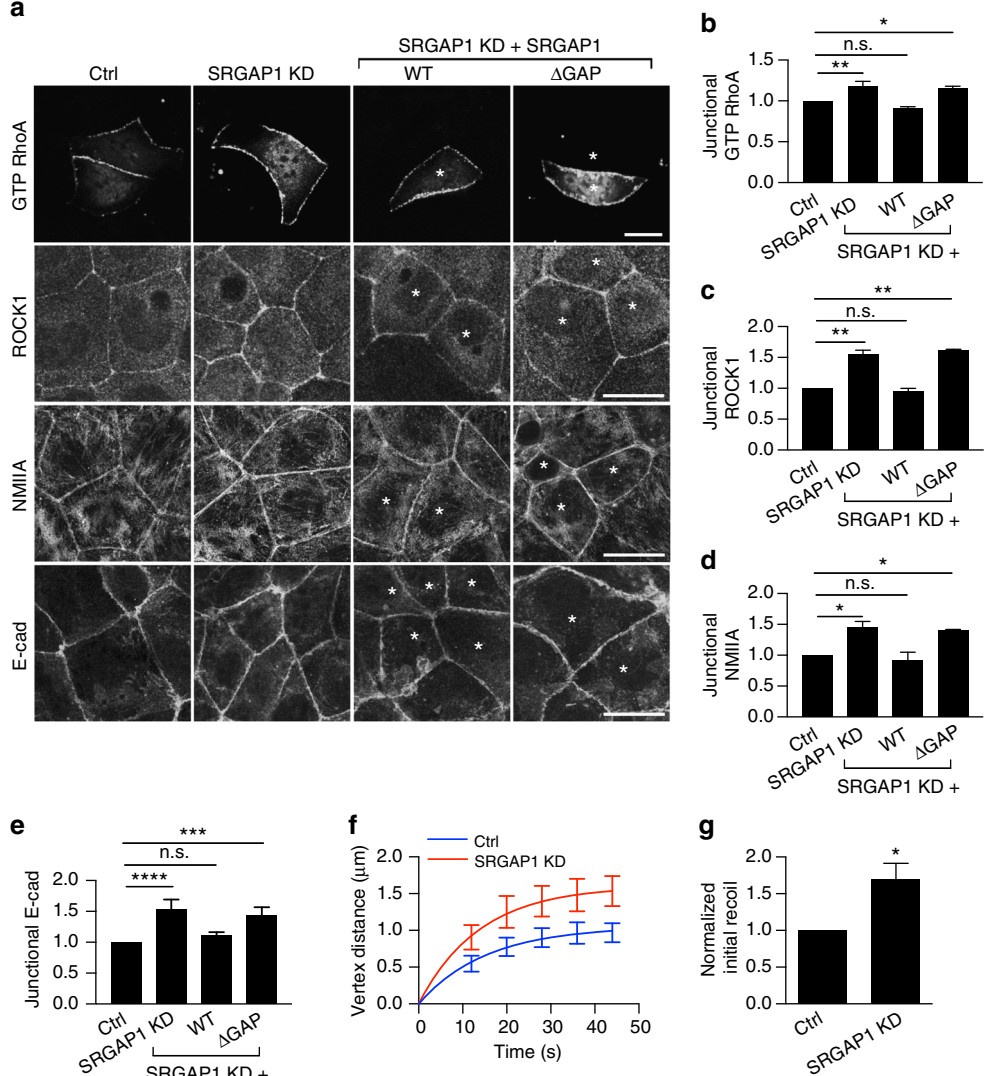

**Fig. 4** SRGAP1 is a RhoA antagonist at the ZA. **a–e** SRGAP1 antagonizes RhoA signaling at the ZA. **a** Representative confocal images of GTP RhoA (assessed by GFP-AHPH), ROCK1, NMIIA and E-cadherin in control (Ctrl), SRGAP1 knockdown (KD), and SRGAP1 KD cells reconstituted with wild-type (WT) or GAP domain-depleted (ΔGAP) SRGAP1. SRGAP1 was depleted by siRNA. *Asterisks* stand for the cells transfected with siRNA and RNAi-resistant SRGAP1 plasmids. Quantification of junctional GTP RhoA (**b**), ROCK1 (**c**), NMIIA (**d**), and E-cadherin (**e**) from the conditions of **a**. **f, g** Recoil measurements (**f**) and initial recoil velocities (**g**) at junctions in control and SRGAP1 KD cells after laser ablation. $N = 3$ independent experiments, data are means ± s.e.m.; one-way ANOVA with Dunnett's post hoc analysis except for **g** where Student's *t*-test was used; n.s., not significant; *$p < 0.05$; **$p < 0.01$; ***$p < 0.001$. *Scale bars* = 15 μm

3YD cortactin caused a decrease in junctional SRGAP1 (Fig. 3h, i). These findings support our biochemical studies (Fig. 3a–e) to suggest that tyrosine non-phosphorylated cortactin can promote the accumulation of SRGAP1 at the ZA, while this is antagonized by tyrosine phosphorylated cortactin.

**SRGAP1 antagonizes the junctional RhoA zone**. As SRGAP1 was detectable at the ZA even in steady-state monolayers, we depleted it by siRNA (Supplementary Fig. 5e–g) to test whether it could affect steady-state junctional RhoA signaling. Indeed, junctional GTP-RhoA was increased by SRGAP1 KD (Fig. 4a, b), suggesting that SRGAP1 acted as a RhoA antagonist. In contrast, we found no changes in junctional GTP-Cdc42 or GTP-Rac1, as measured with location (GFP-RBD-WASP) or FRET activity (Rac-Raichu) biosensors, respectively (Supplementary Fig. 5a–d). Consistent with the observed increase in GTP-RhoA, junctional levels of ROCK1 (Fig. 4a, c), NMIIA (Fig. 4a, d), E-cadherin (Fig. 4a, e), and junctional tension (Fig. 4f, g) were all increased

by SRGAP1 KD. The specificity of these effects for SRGAP1 was supported by using a different siRNA (Supplementary Fig. 5e–g) and confirmed when the junctional changes in GTP-RhoA, ROCK1, NMIIA, and E-cadherin were restored by an RNAi-resistant SRGAP1 transgene (Fig. 4a–e, Supplementary Fig. 4c–e). Of note, however, the effects of SRGAP1 depletion were not restored by the GAP-deleted transgene, although it localized to junctions (Fig. 4a–e, Supplementary Fig. 4c–e), implying that its GAP activity was necessary for SRGAP1 to modulate RhoA signaling at the ZA. Together, these findings suggest that, although in different systems SRGAP1 may also inhibit Cdc42 and Rac1[31, 33, 34], SRGAP1 was an antagonist of RhoA signaling and contractility at the ZA in steady-state Caco-2 monolayers.

These findings raised the possibility that tyrosine non-phosphorylated cortactin downregulated RhoA signaling by enhancing SRGAP1 at junctions. To test this, we asked how 3YF cortactin affected junctional RhoA signaling in SRGAP1 KD cells (Fig. 5). If SRGAP1 mediated the effect of 3YF cortactin, we

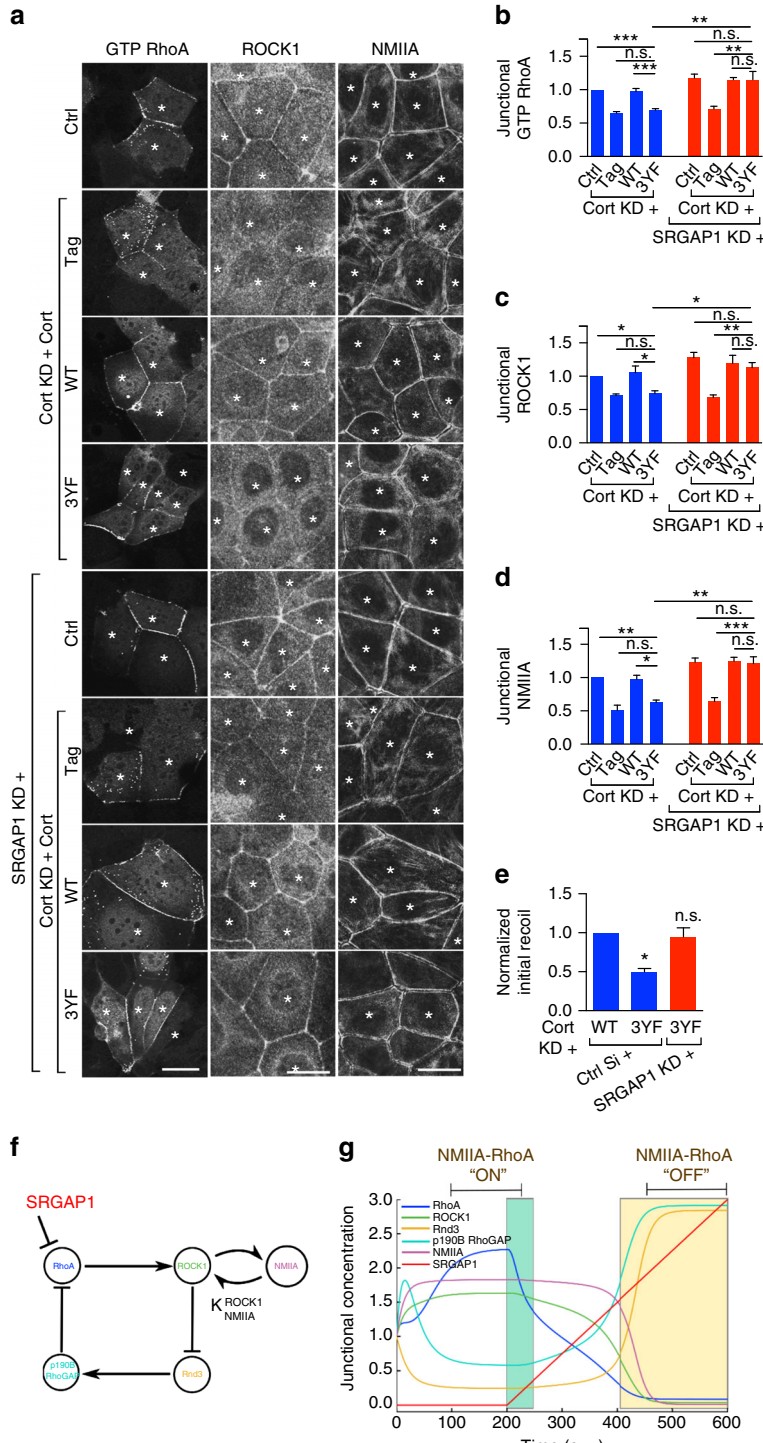

**Fig. 5** Tyrosine non-phosphorylated cortactin downregulates junctional RhoA and contractility via SRGAP1. **a** Representative confocal images of the apical junctional GTP RhoA (assessed by GFP-AHPH), ROCK1, and NMIIA in controls (ctrl) and cortactin shRNA cells (KD) reconstituted with fluorescence-tagged WT or 3YF cortactin (indicated by the *asterisks*). SRGAP1 was depleted by siRNA (SRGAP1 KD). ROCK1 and NMIIA images are maximum projection views of the three apical-most sections. **b–d** Quantification of junctional GTP RhoA (**b**), ROCK1 (**c**), and NMIIA (**d**) from the conditions of **a**. **e** Junctional initial recoil velocities in Caco-2 cells (also see Supplementary Fig. 6c). $N = 3$ independent experiments, data are means ± s.e.m.; one-way ANOVA with Dunnett's post hoc analysis; n.s., not significant; *$p < 0.05$; **$p < 0.01$; ***$p < 0.001$. *Scale bars* = 25 μm. **f** Model of SRGAP1 as an antagonist orthogonal to junctional NMIIA-RhoA feedback network. *Arrows* and *T-junctions* represent stimulation and repression, respectively. **g** The network shown in **f** was mathematically modeled as a system of pairwise stimulators or repressors of junctional localization (Supplementary Information). All junctional concentrations were initialized to one and shown across time except for SRGAP1, which was initialized to 0. At time $t = 200$, a linear increase in the concentration of SRGAP1 (0.0075 per time units) was included in the simulations

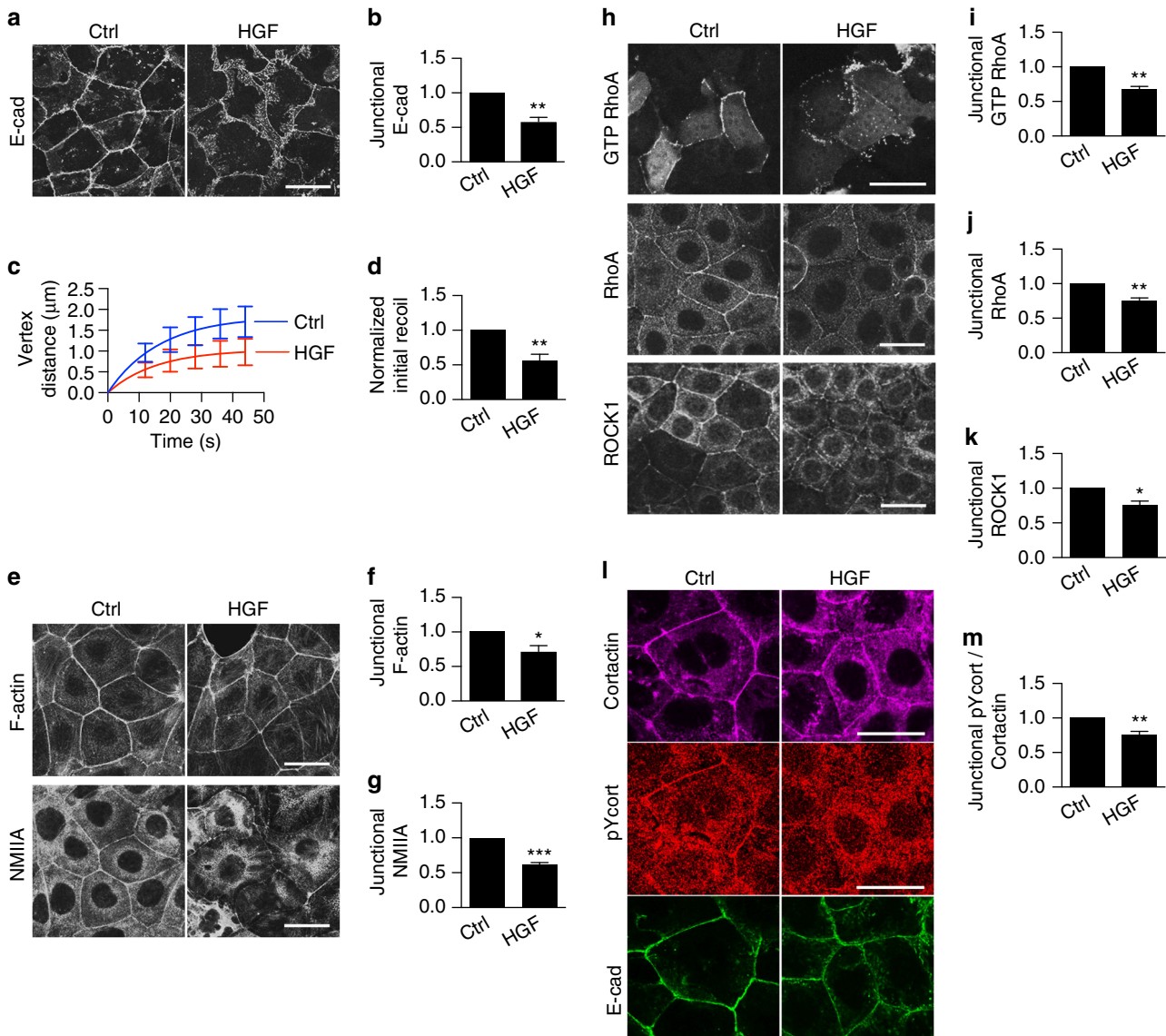

**Fig. 6** HGF downregulates contractility and increases tyrosine-dephosphorylated cortactin at the ZA. **a–g** HGF affects ZA organization and contractility. **a** Representative immunofluorescence confocal images of apical E-cadherin (E-cad) in control (ctrl) and HGF-treated (50 ng ml⁻¹, 16 h) Caco-2 monolayers. **b** Fluorescence intensities of E-cadherin at the ZA in control and HGF-treated cells. **c**, **d** Recoil measurements (**c**) and initial recoil velocities (**d**) of cell–cell junctions after laser ablation in control and HGF-treated cells. **e–g** Representative immunofluorescence confocal images (**e**) and fluorescence intensity of apical F-actin (**f**) and NMIIA (**g**) in control and HGF-treated cells. **h–k** HGF antagonizes the junctional RhoA zone. **h** Distribution of GTP RhoA (assessed with GFP-AHPH), RhoA, and ROCK1 upon HGF treatment. Quantification of junctional GTP RhoA (**i**), RhoA (**j**) and ROCK1 (**k**). **l**, **m** HGF leads to cortactin tyrosine dephosphorylation at the junctions. **l** Representative images of junctional cortactin, tyrosine421-phosphorylated cortactin (pYcort) and E-cad in Caco-2 cells. **m** Fluorescence intensities of pYcort normalized to cortactin at the ZA. $N = 3$ independent experiments; data are means ± s.e.m.; Students' $t$-test; $*p < 0.05$; $**p < 0.01$; $***p < 0.001$. Scale bars = 25 μm

predicted that junctional RhoA signaling would be restored to 3YF cortactin cells by depleting SRGAP1. As noted previously, junctional GTP-RhoA levels monitored with GFP-AHPH were reduced in cortactin KD cells reconstituted with 3YF cortactin (Fig. 5a, b), compared with WT cortactin, conditions that coincided with enhanced levels of SRGAP1. However, junctional GTP-RhoA in 3YF cortactin cells was restored to control levels when SRGAP1 was also depleted (Fig. 5a, b). SRGAP1 siRNA also restored junctional levels of ROCK1 in 3YF cortactin cells (Fig. 5a, c), along with NMIIA content (Fig. 5a, d), E-cadherin (Supplementary Fig. 6a, b) and junctional tension (Fig. 5e, Supplementary Fig. 6c). Overall, these data argue that tyrosine non-phosphorylated cortactin downregulates junctional RhoA signaling through SRGAP1.

**Computational analysis of SRGAP1 in RhoA regulation.** We recently demonstrated that RhoA and NMIIA form part of a mechanochemical signaling network that stabilizes steady-state RhoA signaling at the ZA by inhibiting the junctional localization of the RhoA antagonists, Rnd3 and p190B RhoGAP[29]. Computational modeling revealed that feedback within the network generates bistable outcomes, one state where RhoA was stably active and a second stable state where it was relatively inactive[29]. Our current experiments identified SRGAP1 as a RhoA antagonist that potentially lies orthogonal to the NMII-RhoA feedback network (Fig. 5f). To gain insight into the implications that this architecture might have for RhoA regulation, we then extended our computational model to include SRGAP1. As discussed in the Supplementary Information, this

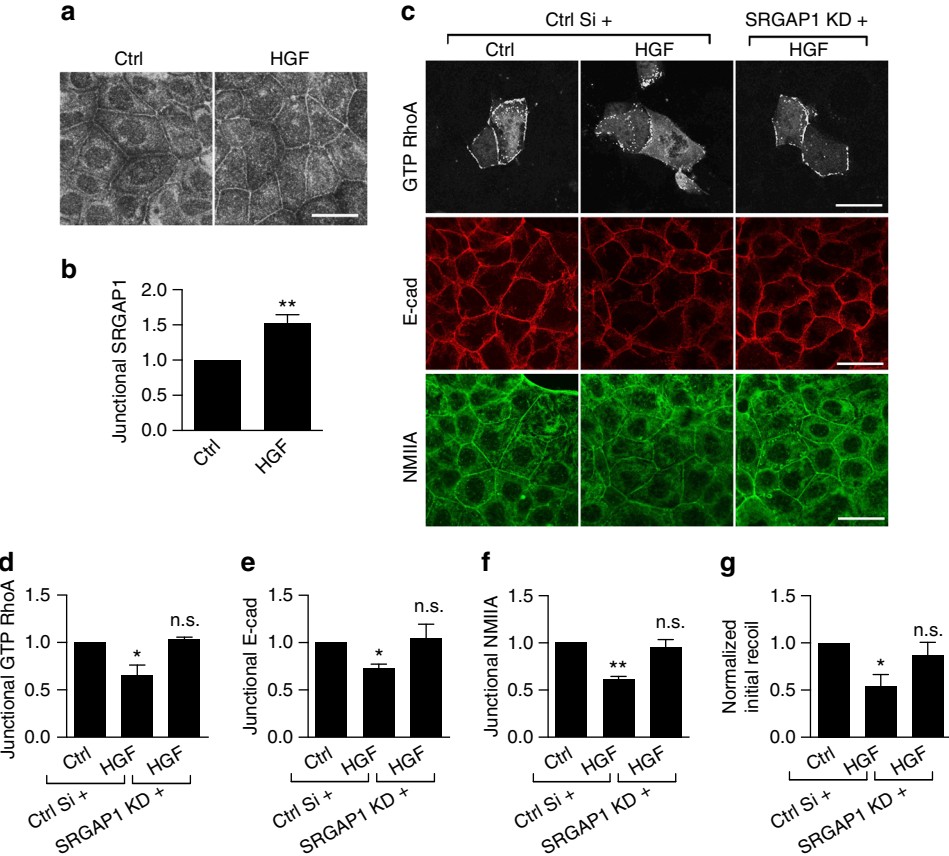

**Fig. 7** HGF disrupts the contractile ZA via SRGAP1. **a**, **b** HGF enhances junctional SRGAP1. Representative immunofluorescence images (**a**) and quantification (**b**) of junctional SRGAP1 in control (ctrl) and HGF-treated cells. $N = 3$ independent experiments, data are means ± s.e.m.; Student's $t$-test; **$p < 0.01$. **c**–**f** Effects of HGF on junctional GTP RhoA (**c**, **d**), E-cadherin (**c**, **e**), and NMIIA (**c**, **f**) in control and SRGAP1 KD cells. **g** Initial recoil velocities in control and SRGAP1 KD cells upon HGF treatment (also see Supplementary Fig. 8d). $N = 3$ independent experiments, data are means ± s.e.m.; one-way ANOVA with Dunnett's post hoc analysis; n.s., not significant; *$p < 0.05$; **$p < 0.01$. Scale bars = 25 μm

model assumes that proteins localized at the junctional cortex are active, including SRGAP1. We began numerical simulations using previously-described conditions that establish a state at the junctions where RhoA and NMII are stably active and Rnd3/p190B are correspondingly suppressed after 100-time units in the simulations[29]. Then we linearly increased the concentration of SRGAP1 and monitored the response of the other components of the network. Interestingly, we found that at low concentrations of SRGAP1 (Fig. 5g, green region), the levels of active RhoA were decreased without being driven fully to the inactive state, in contrast to our previous experience when Rnd3/p190B were manipulated[29]. However, higher concentrations of SRGAP1 then drove the network into the inactive state. This implies that SRGAP1 could regulate RhoA in two regimes: one that occurs at low SRGAP1 levels, where it tunes the NMIIA-RhoA network within the active state (Fig. 5g, green region), and a second that occurs above a threshold of SRGAP1 where the system is driven to inactivate RhoA and NMII (Fig. 5g, orange region).

**HGF inhibits contractility at the ZA**. To then test the potential physiological implications of the SRGAP1 pathway that we had identified, we sought to identify circumstances where junctional cortactin became tyrosine-dephosphorylated. We focused on HGF, which perturbs the actomyosin-dependent ability of cells to concentrate E-cadherin into a ZA[1, 15] when added to confluent epithelial monolayers[9, 35]. We confirmed this in our current experiments (50 ng ml$^{-1}$ HGF, 16 h; Fig. 6a, b; Supplementary

Fig. 7a, b) and further found that HGF decreased junctional tension (Fig. 6c, d), NMIIA, and F-actin (Fig. 6e–g). Thus, HGF downregulated junctional contractility in confluent monolayers.

We therefore wondered whether HGF affected junctional RhoA signaling. Indeed, HGF decreased junctional GTP-RhoA (GFP-AHPH, Fig. 6h, i), TCA-fixed RhoA (Fig. 6h, j) and ROCK1 (Fig. 6h, k). These changes resembled what we had observed in cortactin KD cells expressing 3YF cortactin. Further, HGF decreased junctional pY421-cortactin levels (Fig. 6l, m, corrected for total cortactin protein). Although the amount of cortactin that co-immunoprecipitated with E-cadherin was not substantively altered, the level of pY421 labeling was decreased upon HGF treatment (Supplementary Fig. 7c). Overall, these findings suggested that HGF promoted tyrosine dephosphorylation of cortactin at adherens junctions.

**SRGAP1 mediates the impact of HGF**. As the ability of HGF to downregulate junctional contractility and RhoA signaling correlated with tyrosine dephosphorylation of cortactin, we hypothesized that it might be mediated by SRGAP1. Indeed, HGF increased junctional SRGAP1 staining (Fig. 7a, b) without altering its total expression levels (Supplementary Fig. 8a), but did not promote any evident accumulation of SRGAP1 or cortactin at focal adhesions (Supplementary Fig. 8b, c). Furthermore, SRGAP1 KD prevented HGF from downregulating junctional GTP-RhoA (Fig. 7c, d). This was accompanied by preservation of junctional contractility, as ZA integrity (Fig. 7c, e), junctional

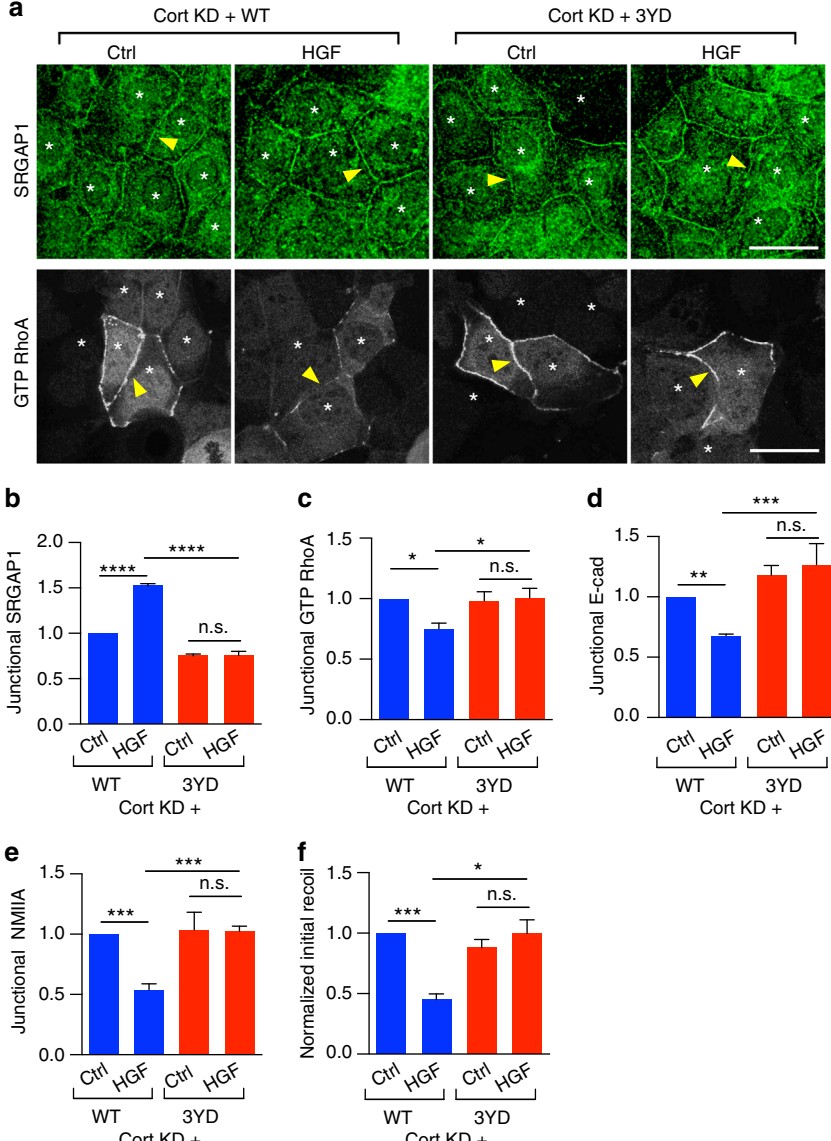

**Fig. 8** HGF recruits SRGAP1 to the ZA via cortactin. Cortactin shRNA Caco-2 cells were reconstituted with WT or 3YD cortactin (marked with *asterisks*) and stimulated with HGF. SRGAP1 images are maximum projections of the three apical-most sections. **a**−**e** Representative confocal images and quantifications of SRGAP1 (**a**, **b**), GTP RhoA (assessed by GFP-AHPH; **a**, **c**), E-cadherin (**d**) and NMIIA (**e**). *Arrowheads* indicate homotypic junctions. **f** Initial recoil velocities in control and HGF-treated Caco-2 cells (also see Supplementary Fig. 9b). $N = 3$ independent experiments, data are means ± s.e.m.; one-way ANOVA with Dunnett's post hoc analysis; n.s., not significant; *$p < 0.05$; **$p < 0.01$; ***$p < 0.001$; ****$p < 0.0001$. *Scale bars* = 25 μm

NMIIA levels (Fig. 7c, f), and junctional recoil (Fig. 7g, Supplementary Fig. 8d) were all rendered resistant to HGF by SRGAP1 KD. Interestingly, reduced junctional F-actin levels were not restored to HGF-treated cells by SRGAP1 KD (Supplementary Fig. 8e, f), which may reflect the ability of HGF to regulate actin anchorage by other signaling pathways[9].

Then, we asked if cortactin mediated the recruitment of SRGAP1 by HGF. As a test of principle, we expressed the 3YD cortactin mutant, which could antagonize SRGAP1 recruitment (Fig. 3h, i), in cortactin KD cells. Compared with control cells expressing WT cortactin at similar levels (Supplementary Fig. 1a, d–f), the junctional recruitment of SRGAP1 by HGF was blocked in cortactin KD cells reconstituted with 3YD cortactin (Fig. 8a, b). Further, we found that reconstitution with 3YD cortactin preserved junctional GTP-RhoA levels (Fig. 8a, c), ZA integrity (Fig. 8d, Supplementary Fig. 9a), junctional NMIIA (Fig. 8e, Supplementary Fig. 9a), and junctional contractility (Fig. 8f,

Supplementary Fig. 9b), despite stimulation with HGF. Overall, we conclude that HGF downregulates junctional contractility by stimulating the junctional recruitment of SRGAP1 to inhibit RhoA signaling.

**SRGAP1 allows HGF to stimulate epithelial cell motility.** Finally, we asked what functional impact might arise from the ability of HGF to downregulate junctional contractility via SRGAP1. We focused on motility of epithelial cells within populations, which can be influenced by intercellular stresses[36]. HGF promotes the dissociation of subconfluent epithelial islands[37, 38]. But to assess its impact in a more physiological context, we tested how HGF might affect the movement of Caco-2 cells grown as confluent monolayers, the circumstances under which we observed that it downregulates junctional contractility. Cells within control monolayers showed low levels of

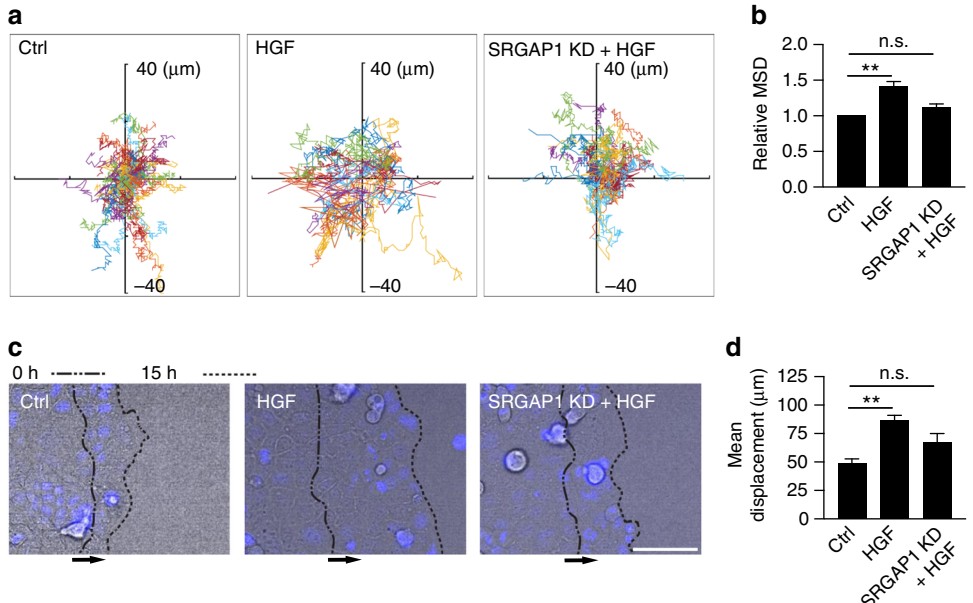

**Fig. 9** SRGAP1 is necessary for HGF-stimulated epithelial cell motility. **a, b** SRGAP1 RNAi perturbs HGF-stimulated intraepithelial cell motility. **a** Representative nuclear tracks (also see Supplementary Movie 1). Thirty to forty nuclei were tracked over 6 h in each movie of the conditions. Axis units: μm. **b** Relative mean squared displacement (MSD) of nuclear movement after 6 h calculated from fitting of Supplementary Fig. 9a. N = 3 independent experiments, 750–1000 tracks were quantitated in each experiment. **c, d** SRGAP1 knockdown perturbs HGF-stimulated wound closure in Caco-2 cell epithelial monolayers. **c** Representative stills of epithelial collective migration from Supplementary Movie 2. *Arrows* indicate the direction of movement. *Dashed lines* indicate the epithelial margins at 0 and 15 h. *Blue color*, nuclei. **d** Mean displacement of the epithelial margins after 15 h. N = 3 independent experiments, 2–4 movies were captured and analyzed in each experiment. Data are means ± s.e.m.; one-way ANOVA with Dunnett's post hoc analysis; n.s., not significant, \*\*p < 0.01. *Scale bar* = 100 μm

locomotility (Fig. 9a, b; Supplementary Fig. 10a) and this was substantially increased by HGF (Fig. 9a, b; Supplementary Fig. 10a and Supplementary Movie 1). However, the ability of HGF to stimulate intraepithelial locomotion was largely abolished by SRGAP1 KD.

We then tested whether this effect might also pertain during epithelial wound healing. HGF stimulated the migration of epithelial sheets into two-dimensional wounds in scratch-healing assays, but this effect was substantially reduced by SRGAP KD (Fig. 9c, d and Supplementary Movie 2). During this process, HGF treatment reduced pY421-cortactin and enhanced SRGAP1 staining at cell–cell contacts (Supplementary Fig. 10b). Cortactin, pY421-cortactin and SRGAP1 were identified in lamellipodia of cells at the migrating front (Supplementary Fig. 10b–g), but their levels were not affected by HGF. Nor was cell-matrix adhesion altered by HGF, SRGAP1 KD or their combination (Supplementary Fig. 10h). Overall, these observations suggested that the role of SRGAP1 in mediating the effects of HGF on epithelial motility might have principally reflected its changes at cell–cell junctions.

## Discussion

RhoA signaling appears as cells assemble junctions with one another[39] and is a prominent feature of the contractile ZA in confluent monolayers[15, 29]. Much effort has been devoted to elucidating how RhoA is activated under these circumstances. It is less clear whether mechanisms exist to actively downregulate junctional RhoA signaling and what functions they may serve. Our current findings now identify such a mechanism, which entails the enhanced junctional recruitment of SRGAP1 by tyrosine non-phosphorylated cortactin. We find that this pathway operates at steady-state cell–cell junctions and is co-opted

by HGF to further relax junctional contractility and promote epithelial cell motility.

First identified during neuronal guidance[31], SRGAP1 has an extensive tissue distribution[33, 40–42], but its function in epithelia is less clear. In the early *C. elegans* embryo, the nematode homolog, SRGP1, localized at epithelial cell–cell junctions, genetically interacted with the cadherin-catenin complex, and promoted the efficient assembly of cell–cell contacts[32]. Our findings extend this to implicate mammalian SRGAP1 in relaxing junctional contractility by downregulating RhoA signaling. Thus, we found that depleting SRGAP1 increased steady-state junctional RhoA signaling, and also increased junctional contractility and cortical actomyosin. In contrast, junctional Rac and Cdc42 signaling were not evidently altered. Further, the ability of SRGAP1 to regulate junctional RhoA signaling required its GAP domain. This implied that the SRGAP1 found at steady-state junctions acted to limit junctional contractility by dampening RhoA signaling.

Multiple mechanisms can recruit SRGAP1 to the cell cortex[32, 33]. Our experiments identified a key regulatory role for tyrosine non-phosphorylated cortactin. This was suggested by the observation that in steady-state monolayers junctional SRGAP1 was enhanced when cortactin was reconstituted with the phospho-deficient 3YF mutant; conversely, SRGAP1 levels were reduced by the 3YD cortactin. Further, we found that in vitro dephosphorylation of cortactin-GFP enhanced its ability to bind SRGAP1; conversely, increasing cortactin tyrosine phosphorylation decreased the association between the two proteins. Consistent with this, ectopically-expressed 3YF cortactin co-immunoprecipitated preferentially with SRGAP1, compared with either WT cortactin or 3YD cortactin. This implies that a direct or indirect biochemical interaction with SRGAP1 may be modulated by the tyrosine phosphorylation status of cortactin.

The precise mechanism remains to be identified, and will depend on understanding how SRGAP1 is recruited to junctions. Different domains of SRGAP1 can influence its subcellular localization in different contexts, including the BAR domain in nematode embryonic epithelia[32] and the GAP domain in fibroblasts[33]. This may also pertain in our experimental model, as SRGAP1 was also seen in lamellipodia of migrating cells. However, redundancy amongst these regions has also been reported in C. elegans[32] and we observed similar results in Caco-2 cells. Therefore, multiple molecular mechanisms may localize SRGAP1 to junctions and resolving these will be an important challenge for future research.

Several lines of evidence indicate that this pathway to recruit SRGAP1 is further stimulated by HGF. Thus, when HGF was added to confluent Caco-2 cells, it decreased actomyosin contractility and RhoA signaling at cell–cell junctions. This coincided with tyrosine dephosphorylation of cortactin and increased junctional SRGAP1. Importantly, the ability of HGF to downregulate RhoA signaling and junctional tension was blocked by SRGAP1 KD. Expression of 3YD cortactin also prevented HGF from increasing SRGAP1 and preserved junctional RhoA signaling and contractility. Together, these observations suggest that HGF may downregulate junctional RhoA signaling by dephosphorylating cortactin to enhance SRGAP1 recruitment. HGF can signal to protein phosphatases, such as SHP-1/2[43, 44] and PTP1B[45], which can also interact with cortactin[46–49], providing potential pathways for HGF to tyrosine-dephosphorylate cortactin.

These findings identify SRGAP1 as a RhoA antagonist at cell–cell junctions, whose capacity to dampen steady-state contractility is co-opted by HGF signaling to further relax junctions. Interestingly, in earlier work we also identified p190B RhoGAP as a potent antagonist of RhoA signaling at the ZA[15]. Whereas SRGAP1 is found at steady-state zonulae adherente, p190B was excluded by mechanochemical feedback from stabilized cortical NMII[29], which anchored ROCK1 at the ZA to antagonize Rnd3, the mechanism responsible for recruiting p190B to the cortex[29]. This feedback network generates bistable outcomes for RhoA signaling[29] and can also define the size and borders of the RhoA signaling zone[50]. However, once NMII-RhoA feedback is established at the mature ZA, its capacity to exclude p190B implied that another GAP might be required to modulate RhoA signaling.

From this perspective, SRGAP1 can be considered as a RhoA antagonist that may act orthogonal to the NMIIA-RhoA network. Indeed, incorporation of SRGAP1 into a computational model of the NMIIA-RhoA network demonstrated that low levels of SRGAP1 could modulate RhoA signaling within a persistent range of activity, but above some threshold SRGAP1 then effectively drove the RhoA zone into a relatively inactive state. We hypothesize that the behavior of steady-state monolayers may correspond to the first regime where junctional GTP-RhoA signaling zone is maintained despite the presence of SRGAP1, whereas HGF acts to drive the system into the second regime. The precise relationship between SRGAP1 and p190B will be an interesting issue for future research. One possibility is that the threshold for SRGAP1 to enter its second regime may reflect when sufficient SRGAP1 has accumulated to overcome buffering by the NMII-RhoA feedback network.

But what function might be served by this pathway for junctional relaxation? Our data suggest that it can facilitate epithelial migration. Thus, SRGAP1 RNAi blocked the ability of HGF to increase both random intraepithelial locomotility within monolayers and collective migration in wound-healing assays. Potentially, cortactin and SRGAP1 may also influence epithelial migration by regulating cell-substrate interactions. Indeed,

although we did not identify it at focal adhesions, cortactin KD reduced cell-matrix adhesion in an apparently Arp2/3-dependent manner. However, cell-matrix adhesion was restored by cortactin phospho-mutants as effectively as it was by WT cortactin; nor did HGF or SRGAP1 KD affect cell-matrix adhesion. Thus changes in junctional contractility associated with tyrosine non-phosphorylation cortactin correlated better with altered migration than did change in cell-matrix adhesion. The notion that changes in ZA contractility may influence epithelial migration is consistent with increasing evidence that macroscopic patterns of tissue stress and fluidity can affect cell movement within cohesive populations[36, 51].

Of note, although HGF can cause subconfluent epithelial islands to disperse[37], this does not occur under more physiological circumstances[52]. For example, cell–cell junctions persist when HGF induces MDCK cysts to rearrange and form tubules[53]. But how such intraepithelial rearrangements occur has been an open question. Our current findings suggest a role for the downregulation of junctional RhoA signaling. As RhoA-dependent junctional actomyosin promotes both cell–cell adhesion and junctional tension, decreased RhoA signaling may then permit cell–cell rearrangements to occur without compromising overall epithelial integrity.

In summary, we have identified a pathway for tyrosine non-phosphorylated cortactin to actively downregulate junctional contractility and potentially facilitate epithelial migration. In this model, cortactin can serve as a phospho-regulated switch that tunes the recruitment of SRGAP1 to zonulae adherente. This provides an alternative perspective to understand how tyrosine phosphorylation of cortactin can promote junctional integrity[21]. Without excluding other potential gain-of-function effects that may occur, we predict that tyrosine phosphorylation would prevent cortactin from recruiting SRGAP1 and thereby protect junctional RhoA signaling. Thus, the positive effect on RhoA signaling that is observed when cortactin becomes tyrosine phosphorylated may actually reflect inhibition of a RhoA-antagonistic pathway.

It should be noted that junctional RhoA signaling was also compromised by cortactin KD alone. However, this effect was not altered by SRGAP1 KD, suggesting that it reflects another pathway for cortactin to influence RhoA. We suggest that this SRGAP1-independent pathway is likely to reflect the ability of cortactin to support actin assembly[17, 18], which helps recruit NMII to junctions[12, 14]. Indeed, we found that W22A cortactin, which cannot support Arp2/3, reduced junctional NMII and tension. As NMII is required to support junctional RhoA signaling[29], its loss in cortactin KD cells could compromise RhoA. Cortactin may then have two contributions to RhoA signaling at the ZA: a basal requirement that reflects its vital contribution to biogenesis of junctional actomyosin, and a regulatory role where its degree of tyrosine de/phosphorylation tunes RhoA signaling through recruitment of SRGAP1. This highlights the complex inter-relationship between actomyosin biogenesis and cell signaling at adherens junctions.

## Methods

**Cell culture**. Caco-2 and HEK293T cell lines were obtained from American Type Culture Collection and were mycoplasma free. For FRAP analyses, E-cadherin in Caco-2 cells was tagged at its C-terminus with enhanced GFP (EGFP). The method and characterization of the cell line are described in detail in Supplementary Information. Caco-2 cells were maintained in RPMI (Gibco) supplemented with 10% FBS, 1% non-essential amino acids (NEAA), 1% L-glutamine, 100 units per ml penicillin and 100 units per ml streptomycin. For HGF (Invitrogen, Cat# PHG0254) treatment, cells were cultured in the above medium to 80% confluency, then switched to a FBS free medium for 1 h and subsequently incubated for 16 h in the same serum free medium supplemented with 50 ng ml$^{-1}$ HGF before fixation. During time-lapse live cell imaging, the cells were treated with HGF in RPMI medium supplemented with 2% serum. Human Embryonic Kidney 293

T cells (HEK293T) were maintained in DMEM (Gibco) medium supplemented with 10% FBS, 1% NEAA, 1% L-glutamine, 100 units per ml penicillin and 100 units per ml streptomycin.

**shRNA, siRNA, and plasmids.** Cortactin transgenes were expressed in cells using a lentivirus-based system[18]. shRNA against human cortactin (5′-GAGAAGCACG AGTCACAGA-3′) was cloned downstream of the U6 promoter within the pLL5.0 vector, which also encodes fluorescent proteins (EGFP or mCherry) as reporter genes whose expression is driven by a LTR promoter[54, 55]. The shRNAi-resistant human cortactin sequence with four silent mutations was cloned downstream of the LTR promoter (BamHI and EcoRI) and upstream of the EGFP or mCherry gene. 3YF (Y421, Y470, Y486), 3YD, W22A cortactin mutant constructs were generated by site-directed mutagenesis of the above vector using the QuikChange Site-Directed Mutagenesis Kit (Stratagene, Cat# 200524). Lentiviral particles were produced in HEK293T cells and were prepared using 100 kDa centrifugal filters (Millipore)[29]. Caco-2 cells were transduced with lentiviral particles for 36–48 h before experiments. Src$^{Y527F}$-mCherry was obtained by cloning the Src$^{Y527F}$ chicken cDNA from Src$^{Y527F}$-GFP (a gift from Margaret Frame, The University of Edinburgh, UK) and cloned into XhoI/BamHI sites in pmCherry-N1 (Clontech).

Human SRGAP1 cDNA was introduced into pEGFP-N1 vector (using XhoI and BamHI). We used two siRNAs (siRNA#1, 5′-UUAACGAUCUGAUUUCUU G-3′[34]; siRNA#2, 5′-CCAGUCCAGGCAGAGCUCAUGCUCA-3′[33]; 50 nM) against the human SRGAP1 that were transiently transfected into cells to deplete SRGAP1 protein expression. siRNA#1 was used through this study except for Fig. 4a–e where siRNA#2 was used. The plasmids encoding myc-tagged siRNA#2-resistant WT, ΔF-BAR-FX, ΔGAP, and ΔSH3 SRGAP1 were kindly provided by Dr. D. Yamazaki (Osaka University, Japan)[33]. The cortactin siRNA was purchased from Invitrogen (CTTN HSS103233). Plasmids and siRNAs were transfected into cells using or Lipofectamine 2000 or RNAi Max reagents (Invitrogen) following the manufacturer's instructions. Cells were analyzed 36–48 h post-transfection.

Caco-2 cells were transfected with 250 ng ml$^{-1}$ GFP-AHPH biosensor, a kind gift from M. Glotzer (University of Chicago, USA) and described previously[29], 36 h before 4% paraformaldehyde (PFA) fixation for confocal microscopy.

**Microscopy and immunofluorescence.** Fluorescent images were acquired on an Olympus IX-81 inverted microscope (×60, 1.40NA Plan Apo objective) with a Hamamatsu Orca-1 ER camera driven by Metamorph imaging software (version 7; Universal Imaging), a Zeiss LSM-510 META confocal microscope (×63, 1.40NA Plan Apo objective), or a Zeiss LSM-710 FCS confocal microscope (×40, 1.30NA; ×63, 1.40NA; ×100, 1.46NA Plan Apo objectives) driven by ZEN software (ZEN 2009; Zeiss). Time-lapse live cell imaging for nuclear tracking and collective wound closure was performed on a Nikon Ti-E deconvolution microscope (×20, 0.75NA; ×40, 0.95NA Plan Apo objectives) driven by NIS-Elements AR software (version 4.3; Nikon) equipped with a 37 °C, 5% CO$_2$ chamber. Panels of images within figures correspond to the same experiment and are representative of three independent experiments unless otherwise specified.

For immunofluorescent staining, confluent monolayers grown on glass coverslips were fixed with 4% PFA in cytoskeleton buffer (10 mM PIPES, 300 mM sucrose, 100 mM NaCl, 3 mM MgCl$_2$, 1 mM EGTA), 10% trichloroacetic acid (TCA) on ice; or methanol at −20 °C for 15 min as indicated below. TCA fixation was terminated by 5 min incubation with 30 mM glycine in phosphate-buffered saline (PBS) and cells were then permeabilized with 0.25% Triton X-100, 5 min at room temperature, then blocked with 5% bovine serum albumin (BSA) in PBS (incubation buffer) and subsequently incubated with primary antibodies (except for pY421 cortactin and SRGAP1) at 4 °C overnight. For pY421 cortactin and SRGAP1 staining, PBS was replaced with Tris-buffered saline (TBS) and the incubation buffer was 0.5% fish gelatin in TBS with 1 mM sodium orthovanadate. After washing, cells were incubated with secondary antibodies at room temperature for 2 h and coverslips mounted on glass slides. E-cadherin, NMIIA, F-actin, cortactin, and pY421 cortactin were stained in PFA fixed cells; ROCK1 and SRGAP1 were stained in methanol fixed cells; RhoA was stained in TCA fixed cells.

**Image processing and analysis.** Quantitative analysis of proteins at the junctions was performed in ImageJ software (NIH) with the line scan functions using lines of 20 μm in length (averaged over 30 pixels) perpendicular to randomly chosen homotypic junctions (where both cells contributing to the junctions were similarly manipulated)[1, 13]. To correct for variation in cell height, optical stacks were acquired which ensured all junctions analyzed were in focus. The Plot Profile feature of ImageJ was used to record the pixel intensities along the selected line. To quantitate E-cadherin, RhoA, ROCK1, SRGAP1, cortactin, the fluorescence profiles were fitted to a Gaussian curve and the peak values (fluorescence intensities at junctions) were obtained from this fitting. The junctional intensity values of pY421 cortactin were identified from intensity profiles for cortactin staining, measured from the regions where cortactin showed its peak intensity values. However, for F-actin and NMIIA, we calculated the area under the curve of the line scan profiles as junctional F-actin and NMIIA often form two-parallel bands alongside the junctions. To calculate the area under the curve, we took pixel intensity values of 10 pixels on both sides from the center of the perijunctional bundles and summed the values. For these analyses, the pixel intensity values lying 10 pixels on either

side of the maxima or centers were considered as background and subtracted from the plots. To correct for fluctuations in the height of cells within frames, some representative confocal images (identified in figure captions) are presented as maximum projection views of the three most-apical sections of the cells (0.19 μm intervals between the sections).

To quantify junctional GTP RhoA from cells that express GFP-AHPH, we measured the ratio of junctional GFP-AHPH and cytoplasmic GFP-AHPH mean fluorescence using the freehand tool built in ImageJ. A 20 pixel wide line was drawn covering the region of whole junctions, and the mean pixel intensity of this region was measured as the junctional GFP-AHPH. Mean cytoplasmic GFP-AHPH fluorescence was measured by masking the cytoplasm of cells and calculating the average pixel intensities within this mask.

**Statistics and repeatability of experiments.** We used standard deviation from initial trials to determine sample size based on confidence interval calculations at confidence levels of 95%. The experiments were not randomized, and we were not blinded to allocation during experiments and outcome assessment. However, a third party with no previous knowledge of the hypothesis aided in quantification of immunofluorescence, and processing the co-immunoprecipitation (Fig. 3) and adhesion assay experiments (Supplementary Figs. 3, 10). At minimum of 25 (IF) or 60 (GFP-AHPH) homotypic junctions from raw images were measured in each experiment. The mean values of different experimental conditions were normalized to the mean value of control conditions in the same experiment. Normalized values were subsequently compared across three independent experiments. Statistical significance of the data was assessed by calculating the P-values using two-tailed Student's t-tests or one-way ANOVA with Dunnett's post hoc test as detailed in the figure legends.

**Two-photon laser ablation to measure junctional tension.** Two-photon laser ablation was performed on a Zeiss LSM-510 META confocal microscope (×63, 1.40NA Plan Apo objective) equipped with a Chameleon multiphoton laser and a 37 °C heating stage[14, 56]. The E-cadherin junctions of Caco-2 cell were labeled by stably expression of shRNA resistant E-cad-GFP in E-cadherin depleted cells via lentiviral infection[57]. A constant region (3.8 × 0.6 μm, with the longer axis orthogonal to the junctions) of the apical junctions was ablated by a femtosecond infrared two-photon 790 nm laser with 20% transmission and 15 iterations. The subsequent retraction of junctional vertices was recorded by time-lapse imaging (6 frames in total; 12 s interval between frame 1 and 2 where the ablation was performed, 8 s interval between rest frames). The distance between vertices of the ablated contacts was tracked in ImageJ and measured as a function of time. The mean values of vertices distance were plotted against time and used for fitting to calculate initial recoil and time constants (Supplementary Methods). At least 12 junctions were ablated and calculated for each condition of every experiment. The calculated initial recoil of each experimental condition was normalized to the initial recoil of the control condition of the same experiment and subsequent values were compared between conditions over three independent experiments.

**FRAP.** FRAP of E-cadherin-GFP was performed to measure E-cadherin dynamics at the ZA[25]. FRAP was performed on a Zeiss LSM-710 FCS confocal microscope (×63, 1.40NA Plan Apo objective) equipped with a 37 °C heating stage. The CRISPS-Cas9 based genome-edited E-Cad-GFP Caco-2 line was used to perform the experiment (Supplementary Information). At each cell–cell contact, E-cad-GFP in a constant circular ROI of 2.7 μm diameter with the center at the contact was bleached with three iterations of the 488 nm laser with 100% transmission. This resulted in a photobleaching of over 70%. Time-lapse images were acquired (three frames) before and after photobleaching (57 frames) with an interval of 5 s per frame. Average fluorescence intensity values $F(t)$ in the bleached area were analyzed with ImageJ. The mean values of the three frames before bleaching was used as the pre-bleached value $F(i)$. The value of the first frame after bleaching was defined as $F(0)$. FRAP values were then calculated as:

$$\text{FRAP} = \frac{F(t) - F(0)}{F(i) - F(0)} = \text{Mf} \cdot \left(1 - e^{\frac{\ln 2 \cdot t}{t_{1/2}}}\right),$$

and plotted over time, where Mf is the mobile fraction, $t_{1/2}$ is the half time of recovery and $t$ is time in seconds. The FRAP values were fitted using a nonlinear regression and the exponential one-phase association model using Y0 = 0 and where Mf corresponds to the plateau value in Prism software[58].

The immobile fraction was then calculated as:

$$\text{immobile fraction} = 1 - \text{Mf} = 1 - \frac{F_\infty - F(0)}{F(i) - F(0)}$$

Six to ten junctions were measured in each experiment. Mean ratios of three independent experiments were plotted and used for statistical analysis.

**Western blotting.** Cells were lysed and the protein samples were collected in lysis buffer [50 mM Tris-HCl, pH 7.4, 2 mM CaCl$_2$, 150 mM NaCl, 0.5% NP40, proteinase inhibitor cocktail (Roche, Cat#138467), phosphatase inhibitor cocktail (Sigma, Cat#P5726)]. Samples were separated by SDS-PAGE and transferred onto

nitrocellulose membranes. After being blocked in 5% non-fat milk in 0.1% Tween 20, TBS, the membranes were incubated with the primary antibody overnight at 4 °C. After washing with TBS containing 0.1% Tween 20, the membranes were incubated with matched horseradish peroxide-conjugated secondary antibody and the signals were developed in enhanced chemiluminescence. All of the uncropped western blots are presented in Supplementary Fig. 12.

**Antibodies**. The primary antibodies used in this study were as follows: mouse monoclonal (mAb) against cortactin (clone 4F11, a kind gift from Scott Weed, West Virginia University, USA; 1:1000 for IF; 1:1000 for Western blotting, WB), rabbit polyclonal (pAb) against tyrosine 421-phosphorylated cortactin (Cell Signaling, Cat#4569S; 1:50 for IF; 1:250 for WB), mouse mAb against E-cadherin (clone HECD-1, a kind gift from Peggy Wheelock, University of Nebraska, USA, with permission from Masatoshi Takeichi; 1:100 for IF, 1:1000 for WB), rabbit mAb against E-cadherin (Cell Signaling, Cat#3195; 1:250 for IF), rat mAb against E-cadherin (Invitrogen, Cat#13-1900; 1:500 for IF), rabbit pAb against GAPDH (Trevigen, Cat#2275; 1:5000 for WB), rabbit pAb against GFP (Invitrogen, Cat#A6455; 1:1000 for WB), mouse mAb against GFP (Molecular probes, Cat# A11120; 1:250 for IF), rabbit pAb against mCherry (Biovision, Cat#5993; 1:500 for WB), mouse mAb against RhoA (Santa Cruz, Cat#SC418; 1:50 for IF; 1:250 for WB), rabbit mAb against ROCK1 (Abcam, Cat#ab143181; 1:100 for IF; 1:500 for WB), mouse mAb against SRGAP1 (Santa Cruz, Cat#SC81939; 1:50 for IF; 1:250 for WB), rabbit pAb against NMIIA (Sigma, Cat#M8064; 1:1000 for IF; 1:1000 for WB), mouse mAb against myc-tag (Cell Signaling, Cat#2276; 1:500 for IF), mouse mAb against β-Catenin (BD transduction, Cat#610154; 1:500 for WB), mouse mAb against α-Catenin (BD transduction, Cat#610194; 1:500 for WB), rabbit pAb against α-Catenin (Invitrogen, Cat#711200; 1:200 for IF), mouse mAb against p120 Catenin (BD transduction, Cat# 610134; 1:500 for WB).

Secondary antibodies for immunofluorescence were species-specific antibodies conjugated to AlexaFluor488, AlexaFluor546 or AlexaFluor647 (Invitrogen). Phalloidin conjugated with AlexaFluor647 or AlexaFluor594 (Invitrogen) was used to label F-actin. The secondary antibodies used for immunoblotting were anti-rabbit or anti-mouse horseradish peroxidase (HRP) conjugated antibodies (Bio-Rad). The secondary antibodies were diluted 1:500 in the experiments.

**Code availability**. Our MATLAB scripts are available from the corresponding authors on request.

**Data availability**. The data that support the findings of this study are available from the corresponding authors on request.

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

## Acknowledgements

We thank all our laboratory colleagues for their support and advice and Dr. Selwin Wu for the initial characterization of the junctional contractile apparatus in response to HGF. This work was supported by grants (1037320, 1067405) and fellowships (1044041 (A.S.Y.), 569542 (R.G.P.), 569512 (M.M.H.), and 1058540 (R.J.D.)) from the National Health and Medical Research Council of Australia, Australian Research Council DP150101367, FT160100366 (G.A.G.)), Queensland Cancer Council grants 1086857, 1128123) and Human Frontiers Science Program grant RGP0023/2014). Optical microscopy was performed at the ACRF/IMB Cancer Biology Imaging Facility, established with the generous support of the Australian Cancer Research Foundation.

## Author contributions

X.L., G.A.G. and A.S.Y. conceived the project and designed the experiments. X.L. performed most of the experiments, in collaboration with G.A.G. for nano-ablation and S.B. for FRAP experiments, with assistance from S.G. for live cells imaging and migration assays, and assistance from S.V. for western blotting and immunoprecipitation experiments. S.B. generated and characterized the CRISPR-based E-Cad-GFP Caco-2 cell line. S.-P.H. generated the plasmids encoding cortactin mutants. G.A.G. and N.A.H. performed the mathematical modeling. M.M.H. processed the samples for mass spectrometry. R.J.D. and R.G.P. contributed scientific inputs and critical comments. X.L. and G.A.G. analyzed the data. X.L., G.A.G. and A.S.Y. wrote the paper.

## Additional information

**Competing interests:** The authors declare no competing financial interests.

**Change history:** A correction to this article has been published and is linked from the HTML version of this paper.

