## [Peer Review File · Nature Communications]

Reviewers' comments:

Reviewer #1, expert in cell-cell adhesions (Remarks to the Author):

This manuscript describes a potential mechanism for loss of contractility during the disassembly of epithelial cell junctions. Liang et al. present evidence in support of the idea that cortactin dephosphorylation disrupts cell junctions by inhibiting RhoA GTP-levels. The mechanism for the loss of RhoA activity involves a recruitment of a RhoA GTP-activating protein known as SRGAP1. Further evidence indicates the HGF usurps SRGAP1 to promote epithelial cell migration.

How epithelial cell contractility is inhibited is an important area of investigation. The ideas presented in this manuscript are intriguing and interesting. However, the data is too preliminary and only somewhat supportive of the authors' conclusion. Along these lines, the major areas of concern are:

(1) There are concerns related to the re-expression of the mutant cortactins. The cortactin staining should be shown alongside the images. What is the level of re-expression? Do the mutant proteins localize to adherens junctions?

(2) Cortactin localizes to integrin-containing sites in the cells, yet the authors attribute their effects to cortactin at adherens junctions without data to support a lack of an effect on cell adhesion to the matrix. This possibility warrants consideration.

(3) The data is not fully convincing. In numerous images, the authors show fields of cells where some of the cells are transfected and others are not. It appears in many instances that the untransfected cell have the same staining pattern as the transfected cells, thereby decreasing confidence in the findings. For example, the cells in Fig 1c with cortactin KD and Cort KD+WT Cort, the cells without asterisks are similar to those with asterisks. Some biochemical measure of E-cadherin levels in adherens junctions is needed to validate the imaging. Similarly, in Figure 4a, the GTP-loading of RhoA imaging is neither robust nor convincing. Do biochemical assays of GTP-bound RhoA support these and the other RhoA-GTP imaging studies?

(4) The functional consequences of the GAP protein on cell migration could be due to a direct effect on integrin-related process and therefore cannot be attributed solely to an effect on contractility at adherens junctions.

(5) The proposed interaction between dephosphorylated cortactin and SRGAP1 is an interesting possibility that needs more exploration. Is cortactin phosphorylation lost during junctional assembly? Do stimuli that trigger a loss of cortactin phosphorylation promotes its association with SRGAP1.

Reviewer #2, expert in cell adhesion, HGF (Remarks to the Author):

The manuscript by Liang and colleagues uncovers how cortactin regulates junctional Rho signaling through recruitment of SRGAP1 and how this mechanism is used by HGF to induce junction-loosening. The results of this study are timely and relevant for a broad audience of (tumor-) cell biologists. As the authors nicely conclude, however, the inter-relationship between actomyosin biogenesis and cell signaling at adherens junctions is complex and several remaining questions/concerns remain about the mechanisms and the direct involvement of this pathway in junction regulation, that would have to be dealt with before publication in Nature Communications.

General comments:

The model contains a circular argument: In cortactin and SRGAP KD cells, there is less Rho activity at/near junctions. However, there is also less E-cadherin, which by itself would lead to less junctional proteins including actin (and cortactin and SRGAP and many others). It can therefore not be concluded that there is a direct effect of cortactin and SRGAP on Rho activity. It could be a general effect on actomyosin structure/organization which then automatically leads to altered/loosened junctions and subsequent. Both cortactin (heavily) and SRGAP (3) have been implicated in lamellipodial protrusion and integrin-dependent cell migration and it is well established that crosstalk through actomyosin organization/contractility exists between integrin and cadherin adhesions. The authors could counter this argument (at least to a decent extent) by directly testing whether cortactin a (and mutants) and SRGAP1 localize to cell matrix adhesion structures and protrusive cell areas and whether their manipulations affect cell migratory processes (they do actually in figure 8, but simply blame the effects on cadherin adhesion with cannot be a priori concluded).

Related to this: SRGAP1 is a GAP, but seems to affect RhoA total protein levels rather than specifically RhoA GTP levels at junctions. This is not at all mentioned or discussed and adds to my skepticism towards the models presented. I have not re-read the modeling paper referred to in figure 4 f,g, but here it should definitely and explicitly made clear how GTPase activating function is incorporated and leads to total reduction rather than loss of activation.

Technically, the manuscript relies on a lot of quantifications of (Immuno-) Fluorescent images in fixed cells. Not all of these images look of sufficient quality to warrant the current use of this technique which has significant complications because it is difficult to define an internal standard as background and real signal do not fluctuate at the same ratio across experiments or even across slides, or even within slides. For instance in several images in figure 2,3,4,5 and 7 the staining do not look very homogeneous and there is no clear reason think that this is due to a biological reason rather than a technical one. Also, the SRGAP1 levels in junctions in the supplemental figure 3B look much less convincing than those in main figure 3e. The authors could relieve some concern here by performing co-stainings and quantifications with a regular junction marker like E-cadherin itself or b-catenin. Ample suitable antibodies are available.

I do not understand why the F-actin levels are not restored by SRGAP1 knockdown in HGF-stimulated cells in supplemental figure 6C. If the modeling is correct, how is this explainable?

In figure 7 I miss the experiments where the effect of cortactin KD (no rescue) on HGF effects and the effect of cortactin YF rescue.

In general, in all the KD rescue experiments, the expression controls are lacking, but they are essential to properly judge the experiment. (as KD , lack of KD and rescues may result in the same effects)

Reviewer #3, expert in GTPases (Remarks to the Author):

The manuscript by Liang and colleagues describes a potential function of SRGAP on the negative regulation of RhoA activity at junctions. This is an important issue on junction regulation that is not yet fully understood: how dynamic cycles of activation and inactivation of small GTPases are controlled spatially at the molecular level. The authors propose key role of cortactin tyrosine phosphorylation to localize SRGAP at junctions, and thereby modulate RhoA activation levels locally. The authors use a series of quantitative assays to demonstrate the impact on junction contractility, localization of different markers at junctions and the ability of cells to scatter in response to different stimuli. Appropriate controls are shown by using different mutants and

rescue experiments. The figures have very nice presentation and are well structured; the text is very clear and readily understood.

Experimentally the work is excellent and the data convincing. However, this reviewer finds conceptual and technical issues that would preclude its publication in its current form. First, conceptually, the distinction between recruitment to junctions and activation has not been properly addressed throughout. This is an important issue as there is clear evidence that many of the RhoA regulators localized at junctions are inactive. Thus recruitment does not always correlate with activation. Second, there are a number of other interpretations to explain the results that were not formally excluded by the authors. These are not tested or discussed in the text. Third, there are concerns with the phospho-mimetic mutant.

Specific points:

1 - SRGAP1 is a GAP for RhoA, Rac1 and Cdc42 in vitro. In cells and depending on the cellular event, GAPs may have a preference for a particular substrate. However, the authors have not excluded the involvement of Rac1 or Cdc42 in their analysis on junctions or indeed motility. In particular Rac1 can control also contractility, E-cadherin stability at junctions and down-regulate RhoA activation via different mechanisms.

2 - The authors equate recruitment to junctions to activation of a pathway. This is not strictly correct and bypasses important concepts and knowledge of the activation status of signalling molecules at junctions. Interpretation of the data should be revised. For example, if the total level of RhoA protein is increased at junctions and its activation is also increased proportionally (Fig.3i,j), it is difficult to dissect which of these two events is being controlled.

3 - The model proposed only uses concentration of a particular molecule, without considering spatial information or activation steps. As presented, the biological impact of such model is reduced. For signalling proteins, the net number of molecules is not relevant, but rather the extent and duration of its activation locally. If inclusion of these additional parameters is not possible in the model, it should be removed or at least these limitations should be raised and discussed in the text.

4 - The use of the tyrosine phosphomimetic mutation as aspartic acid on cortactin is not appropriate. Aspartic acid is much smaller than the bulky tyrosine (or phosphotyrosine) residue and Aspartic acid (or glutamic) is normally used to produce phosphomimetic mutants at Serine or Threonine residues. The impact of the YD cortactin mutant binding to SRGAP1 is not surprising and could be due to structural changes on cortactin protein. Unless more validation is provided, this reviewer is not convinced that it can be interpreted as a phosphomimetic mutant. As this piece of evidence is essential to the conclusions of the manuscript, binding of phosphocortactin to SRGAP1 should be validated by in vitro phosphorylation or de-phosphorylation of WT cortactin and other known in vivo phospho-cortactin functions should be tested with the YF and YD mutants.

5 - The data using anti-pY421 cortactin shown in Fig 5 should be interpreted carefully. The anti-phospho antibody will only bind to freely available epitopes (i.e. those not masked by any interacting proteins). If SRGAP1 binds more efficiently to phosphorylated cortactin (or indeed any other associated protein to phospho-cortactin), then one can envisage that there would be less staining with the anti-phospho antibody.

6 - The localization of active RhoA is very clearly modulated under the different conditions tested. However, the reduction in junctional active RhoA could be due to indirect events. For instance, perturbation of junctional contraction by cortactin perturbations could indirectly modulate the pool of p190RhoGAP as shown in the previous papers from the lab. This possibility was not discussed or eliminated. It would be good to have in the text a broader view of how these two pathways that control RhoA can be coordinated and coexist in the same space.

7 - The use of cell motility to confirm a potential effect of SRGAP1 on junction relaxation is problematic. It has not been shown in the cellular system presented whether SRGAP1 interferes with focal adhesion turnover or motility per se. Without these controls it is difficult to conclude that the motility defects are caused to junction stability and contractility.

Reviewer #4, expert in cortactin (Remarks to the Author):

In this manuscript by Liang and Yap, the authors present evidence that cortactin dephosphorylation recruits the RhoA antagonist srGAP1 to epithelial zonula adherens junctions to downregulate contractility. The authors demonstrate that nonphosphorylatable cortactin (3YF) decreases contractility and RhoA activity at cell junctions by preferentially recruiting srGAP1. Hepatocyte growth factor stimulates cell motility by decreasing cortactin phosphorylation and recruiting srGAP1 to decrease contractility.

This work represents a potential advance in understanding the complementary but distinct roles of cortactin in actomyosin biogenesis and RhoA regulation at cell junctions. However, in some places the data lacks a degree of experimental rigor necessary to support the conclusions drawn by the authors. Thus, without significant revisions it does not currently meet the standards of Nature Communications.

Major Points:

1. The expression levels of cortactin mutants in knockdown cells (shown in Fig S1a, b) is not at the level of endogenous protein. The mCherry-fusions expressed in some cases are below the level of residual endogenous cortactin after knockdown. Given the dependence of the conclusions in Figures 1, 2, 3, 4, and 7 on the expression of cortactin mutants, experiments may need to be repeated with more robust controls for mutant expression relative to wild-type levels (and in excess of residual endogenous protein).
2. Methods for quantification of E-cadherin, RhoA, ROCK1, srGAP1, cortactin, and pY-cortactin are dependent on a peak intensity value based on a line trace across a used-defined junction on cells. While Western blotting confirms expression levels of their proteins are unchanged during perturbations, the method for quantification of GFP-AHPH, using the ratio of mean junction fluorescence relative to mean cytoplasmic fluorescence, would be more rigorous in quantifying the enrichment of these proteins at junctions.
3. What are the details of the interaction screen that reveals srGAP1 as an interaction partner of cortactin-3YF? The authors state that this interaction could be direct or indirect, a hypothesis that is testable with recombinant proteins. Recombinant proteins would also allow for testing dephosphorylated cortactin (incubated with a phosphatase) as opposed to nonphosphorylatable cortactin-3YF.
4. IF the interaction with srGAP1 is direct, the binding site on cortactin is contained within what is likely to be an extended unstructured region. Which regions of srGAP1 bind to cortactin? Do finer point mutations in srGAP1 that disrupt cortactin binding affect srGAP1 junctional recruitment, Rho activity, and junctional recoil?
5. What degree of knockdown is achieved with SRGAP siRNA? Can the blot in Figure S3A or the images in S3B be quantified? While the use of two different siRNAs is indicative of specificity, rescuing the SRGAP knockdown cells with a siRNA-resistant mutant is the gold standard in confirming specificity, as the authors have done for their cortactin knockdown experiments.
6. The authors refer to cortactin dephosphorylation throughout the manuscript, making inferences using phosphomimetic or nonphosphorylatable mutants. As they mention in their discussion, screening for potential phosphatases that regulate cortactin's effects in cell junctions would strengthen this argument considerably and demonstrate that the phenomena observed are due to enzymatic dephosphorylation that occurs in physiological conditions. What is known about the mechanism of dephosphorylation in response to HGF treatment?

7. It would seem important for the authors to confirm that their pY421 antibody is phospho-specific? This could be done testing for signal in cortactin-3YF cell lines using both total cortactin and phospho-specific antibodies. The concern is that residual cortactin immunoreactivity could be contributing to the signal.

8. Figure 8A and Supplementary Movie 1 have a significant confounding factor in the amount of cell division occurring in the HGF conditions. Some of the longest motility tracks seen in the HGF condition are in the case of a cell that divides and rapidly splits. This data should be quantified with those cells expressly excluded.

9. In their conclusion, the authors suggest that srGAP1 is present at cell junctions at a low concentration range that their model indicates will modulate RhoA activity without crossing the threshold to reach an inactive RhoA state. Can the authors, by use of quantitative immunofluorescence in conjunction with immunoblotting, estimate the relative concentrations of srGAP1, cortactin, and RhoA at cell junctions to validate this model?

10. The authors suggest that the W22A mutant of cortactin inhibits cortactin interaction with the Arp2/3 complex. It is possible that cortactin can indirectly recruit and activate the Arp2/3 complex in cells, perhaps through Nck1:N-WASp binding to the cortactin proline-rich region. It would seem imperative for the authors to quantify Arp2/3 localization to cell junctions in each of their cortactin rescue cell lines?

Minor Points:

1. The authors suggest that cortactin is orthogonal to their RhoA regulatory network outlined in Figure 4F. Have they tested for cortactin knockdown effects on p190B or Rnd3 expression and localization? This may be especially relevant in the context of Binamé et al, JBC 2016 showing an interaction between cortactin and p190A.

2. Two typos are present:

a. Downregulation is one word in the manuscript title but two words in the title of Figure 5.

b. The symbol for viscosity coefficient is mu in the equations but eta in the text of the Supplemental Section "Two-photon laser ablation and tension measurements"

RESPONSE TO REVIEWERS

Reviewer #1, expert in cell-cell adhesions

This manuscript describes a potential mechanism for loss of contractility during the disassembly of epithelial cell junctions. Liang et al. present evidence in support of the idea that cortactin dephosphorylation disrupts cell junctions by inhibiting RhoA GTP-levels. The mechanism for the loss of RhoA activity involves a recruitment of a RhoA GTP-activating protein known as SRGAP1. Further evidence indicates the HGF usurps SRGAP1 to promote epithelial cell migration.

How epithelial cell contractility is inhibited is an important area of investigation. The ideas presented in this manuscript are intriguing and interesting. However, the data is too preliminary and only somewhat supportive of the authors' conclusion. Along these lines, the major areas of concern are:

(1) *There are concerns related to the re-expression of the mutant cortactins. The cortactin staining should be shown alongside the images. What is the level of re-expression? Do the mutant proteins localize to adherens junctions?*

RESPONSE: The reviewer makes a very good point. In particular, as the reviewer implies, a key issue is the level to which the transgenes are expressed at adherens junctions, especially relative to the amount of cortactin that is normally found there. In the original submission, we showed that the cortactin transgenes used in this study (WT, 3YF- and 3YD-cortactin) all localize with E-cadherin at adherens junctions to a similar degree (Fig S1d, f) and also by western analysis that these transgenes were expressed to similar extents overall (Fig S1a).

To characterize the degree of reconstitution at junctions, we have now extended this analysis further to test if the transgenes restore junctional cortactin levels to approach that seen in control cells where cortactin was not depleted. To examine the level of re-expression at the junctions we have now co-stained samples for cortactin (using an antibody that recognizes the transgenes as well as endogenous cortactin), reasoning that this would also allow us to assess the extent to which reconstitution approached the levels of cortactin found in control cells. We found all the cortactin transgenes restore cortactin staining at junctions to the same level as seen for endogenous cortactin in control cells transfected with GFP alone. These data (presented in Fig S1d,e) indicate that in our experiments the cortactin transgenes are expressed at similar levels and restore junctional cortactin to control levels.

(2) *Cortactin localizes to integrin-containing sites in the cells, yet the authors attribute their effects to cortactin at adherens junctions without data to support a lack of an effect on cell adhesion to the matrix. This possibility warrants consideration.*

RESPONSE: We agree that this is an interesting area that was overlooked in the original submission of our paper. In particular, we found that cell movement within confluent monolayers, as well as collective epithelial migration into wounds, were increased by HGF and this effect was reduced by SRGAP1 RNAi. We attributed this to changes in junctional tension, consistent with growing evidence (e.g. from the Trepatt group) that cell-cell forces influence collective migration. It is absolutely true that changes in cell-matrix interactions or in the locomotor apparatus might also have occurred. We have sought to pursue this, as suggested by the reviewer, in this revision.

As these experiments focused on epithelial cell migration promoted by HGF, we

focused especially on this context. However, as shown in Fig S7b,c, we could not identify co-localization of either SRGAP1 or cortactin with focal adhesions in either control or HGF-treated monolayers, in contrast to the clear increase in junctional SRGAP1 that was observed. Furthermore, although we do see cortactin, pY-cortactin and SRGAP1 at the leading edges of motile lamellipodia (most evidently at the migrating front during scratch wound assays), this was not altered by HGF (Fig S9 b,c). Therefore, it is not immediately apparent that they were found at cell-matrix adhesions or were substantially altered by HGF. Of course, it remains possible that changes in junctional RhoA may have had secondary effects on cell-substrate interactions by other pathways. However, respectfully, we feel that these are beyond the scope of the present study. However, we have modified the Discussion to make it clear that, in addition to the impact of cell-cell junctions that we favour as an interpretation, we cannot exclude direct or indirect changes to cell-substrate adhesion or locomotility:

“The relationship between junctional relaxation and epithelial motility that we have observed is consistent with increasing evidence that macroscopic patterns of tissue stress and fluidity can influence cell movement within cohesive populations^{34, 49}. Potentially, cortactin and SRGAP1 may also influence epithelial migration by regulating cell-substrate interactions or their dynamics. However, we did not identify either protein at focal adhesions in the epithelial monolayers that we studied, although indirect effects from changes in AJ or junctional signaling cannot be ruled out.”

(3) *The data is not fully convincing. In numerous images, the authors show fields of cells where some of the cells are transfected and others are not. It appears in many instances that the untransfected cell have the same staining pattern as the transfected cells, thereby decreasing confidence in the findings. For example, the cells in Fig 1c with cortactin KD and Cort KD+WT Cort, the cells without asterisks are similar to those with asterisks. Some biochemical measure of E-cadherin levels in adherens junctions is needed to validate the imaging. Similarly, in Figure 4a, the GTP-loading of RhoA imaging is neither robust nor convincing. Do biochemical assays of GTP-bound RhoA support these and the other RhoA-GTP imaging studies?*

RESPONSE: We have endeavoured to improve the quality of the images presented. We agree that the data quality in the submitted PDFs was not as good as it could have been. In particular, one reason for the apparent variability was due to variation in cell height. Our cells do not grow to a uniform height and the apical plane, including the apically-located ZA, can go in and out of focus of a single optical plane between cells that are within a few diameters of one another. One limitation of the previously presented images was that they were taken from single optical planes. Accordingly, we have now:

- a) Improved the contrast handling of the images (but with handling applied uniformly to all images presented).
- b) Compensated for variation in cell height. Whereas we earlier presented single confocal optical slices, we now present maximum-projection views of the apical-most 3 optical slices to better capture the apical region.

We should emphasize, however, that the quantitative analysis was designed to overcome potential sources of variability.

First, to compensate for the problem of variation in cell height, we worked from image stacks and manually measured fluorescence intensity at the apical junctional region by line scan analysis, with the lines oriented perpendicular to randomly chosen junctions. This ensured that we were dealing with apical regions that were in the optical plane of analysis. We have modified the Methods to make this clear.

Second, we always studied the junctions between two comparable cells (effectively homotypic cell-cell contacts) – e.g. between KD cells or KD cells that are both reconstituted with transgenes. This was to avoid possible variability associated with heterotypic contacts. Again, we have modified the methods to make this clear.

Third, the data presented are biological replicates (typically from 3 independent experiments). However, each individual biological replicate was the average of at least 25 individual junctions (technical replicates) for most markers that we measured (and at least 60 junctions for GFP-AHPH experiments). Therefore, the final data presented represent at least 75 individual junctions for most markers (and at least 180 junctions for GFP-AHPH). Therefore, we are confident that the quantitative analysis robustly captures the changes in our experiments.

In addition, we have used a number of proxies, either as markers of effective RhoA signaling (i.e. TCA-resistant RhoA staining at junctions, as we earlier showed that the localization of RhoA at junctions required that it be active; Ratheesh et al., NCB 2012; and ROCK1, whose localization is also sensitive to junctional RhoA, Priya et al., NCB 2015) and downstream markers of contractility (e.g. NMII staining) that we earlier found to also require junctional RhoA signaling (Smutny et al., NCB 2010; Ratheesh et al., NCB 2012). These were altered in a fashion consistent with the observed changes in e.g. GTP-RhoA, as measured with the AHPH biosensor.

Overall, we believe that the analysis of junctional proteins by IF is the best available technique. In contrast, we are not aware of any reliable biochemical technique to measure changes at the ZA. We note here that we use the “ZA” to specifically refer to the apical ring of E-cadherin (and its associated actomyosin cortex), which is the dominant site of contractile tension in these cells (Wu et al., NCB 2014). However, there is plentiful E-cadherin elsewhere, decorating the lateral junctions below the ZA (Wu et al., NCB 2014). To date, we do not have any robust biochemical way of distinguishing between these two pools. For example, detergent-solubility does not reliably distinguish them in our hands, nor does it eliminate the capacity for DRM/raft-like domains to be contributing. Similarly, in our experience RhoA-pull-down assays do not robustly discriminate the pool of the ZA, which is probably small but striking because it is much more stable than many other pools that we have yet found in confluent interphase epithelial monolayers (Priya et al., NCB 2015).

(4) The functional consequences of the GAP protein on cell migration could be due to a direct effect on integrin-related process and therefore cannot be attributed solely to an effect on contractility at adherens junctions.

RESPONSE: Please see response to (2), above. We agree that this is formally possible, but have not found localization of SRGAP1 or cortactin to cell-substrate adhesions, either in control or HGF-treated cells. We favour a role for contractility at cell-cell junctions being a mechanism for SRGAP1 to influence cell migration of, and within, cohesive epithelial collectives. However, as noted above, we have adjusted our discussion to note that an additional effect on cell-substrate interactions cannot be excluded.

(5) The proposed interaction between dephosphorylated cortactin and SRGAP1 is an interesting possibility that needs more exploration. Is cortactin phosphorylation lost during junctional assembly? Do stimuli that trigger a loss of cortactin phosphorylation promotes its association with SRGAP1.

RESPONSE: Indeed, in pilot studies we have observed that the junctional content of pY(421) cortactin increases as junctions assemble. This correlates with an increase in

junctional contractility that is RhoA-dependent. This is what would be predicted if tyrosine phosphorylation of cortactin protects junctional RhoA signaling by decreasing the amount of SRGAP1 that is present at the junctions. We show representative data here for the reviewer (see below). However, we would prefer to keep this data separate from the present manuscript (which is already long and would lose focus with addition of this material).

Indeed, the current focus of this manuscript addresses the reviewer's final question. HGF is, as we show, a stimulus that decreases cortactin tyrosine phosphorylation at the ZA to enhance junctional SRGAP1. Furthermore, in new experiments that we performed (Fig 3c-e) we now show that the *in vitro* dephosphorylation of cortactin (by incubation with λ -phosphatase) decreases the amount of SRGAP1 that associates with it. Conversely, increasing cortactin tyrosine phosphorylation by co-expression with constitutively-active Src decreased the amount of SRGAP1 that associated with isolated cortactin. This reinforces our interpretation that SRGAP1 is preferentially recruited to junctions, directly or indirectly, by the tyrosine de-phosphorylated form of cortactin, while tyrosine phosphorylation can antagonize this effect.

(A and B) Records of vertex separation (A) and initial recoil (B) of apical E-cadherin junctions in Caco-2 monolayers at different confluencies. (C and D) Junctional levels of cortactin (C) and tyrosine 421 phosphorylated cortactin (pY421 cort, D) normalized to cortactin levels in monolayers of different confluencies. (E and F) Junctional GTP RhoA (GFP-AHPH, E) and SRGAP1 (F) in monolayers of different confluencies.

Reviewer #2, expert in cell adhesion, HGF

The manuscript by Liang and colleagues uncovers how cortactin regulates junctional Rho signaling through recruitment of SRGAP1 and how this mechanism is used by HGF to induce junction-loosening. The results of this study are timely and relevant for a broad audience of (tumor-) cell biologists. As the authors nicely conclude, however, the inter-relationship between actomyosin biogenesis and cell signaling at adherens junctions is complex and several remaining questions/concerns remain about the mechanisms and the direct involvement of this pathway in junction regulation, that would have to be dealt with before publication in Nature Communications.

General comments:

1) *The model contains a circular argument: In cortactin and SRGAP KD cells, there is less Rho activity at/near junctions. However, there is also less E-cadherin, which by itself would lead to less junctional proteins including actin (and cortactin and SRGAP and many others). It can therefore not be concluded that there is a direct effect of cortactin and SRGAP on Rho activity. It could be a general effect on actomyosin structure/organization which then automatically leads to altered/loosened junctions and subsequent.*

RESPONSE: Our interpretation in the original manuscript was based on the strong links between the effects of independently manipulating cortactin, SRGAP1 and our earlier experience with RhoA. Thus, reconstitution of cortactin KD cells with 3YF cortactin was associated with both persistently reduced junctional RhoA signaling and an increase in junctional SRGAP1. The decreases in E-cadherin that we observed were also consistent with what we earlier found when RhoA/ROCK signaling is directly blocked (Smutny et al. Nature Cell Biol 2010, 12, 696-702; Ratheesh et al., Nature Cell Biol 2012, 14, 818-828). Furthermore, the fact that depleting SRGAP1 could counteract the impact of 3YF cortactin suggested that the ability of SRGAP1 to downregulate junctional RhoA signaling might be key. It is certainly true that there is extensive feedback at junctions between adhesion, the actomyosin cortex and RhoA signaling (as we have begun to explore: Priya et al., NCB 2015; PLoS Computational Biol, 2017) and this certainly carries analytical challenges.

Therefore, to pursue the hypothesis that downregulation of RhoA by SRGAP1 contributes to its impact at junctions, we have now performed further experiments. We measured junctional RhoA signaling when SRGAP1 KD cells are reconstituted with either WT SRGAP1 or a mutant lacking the GAP domain (Fig 4a,b). As previously, we find that junctional RhoA is increased in SRGAP1 KD cells, accompanied by an increase in junctional ROCK1, NMIIA and E-cadherin (Fig. 4 c,d,e). This increase does not occur when KD cells are reconstituted with WT SRGAP1, confirming that these effects are specific for manipulation of SRGAP1 expression. However, the increase in RhoA signaling persisted when KD cells were reconstituted with the GAP-deleted mutant. Therefore, the GAP domain and, by implication its known RhoA-inactivating capacity, is necessary for SRGAP1 to regulate junctional RhoA (as we also showed that SRGAP1 does not affect junctional Rac1 or Cdc42 signaling, Fig S4a-d). We believe that this new finding reinforces the conclusion that down-regulation of RhoA plays an important role in allowing SRGAP1 to dampen contractility at cadherin junctions. We thank the reviewer for encouraging us to pursue this question.

As a point of clarification, we would note that the impact of cortactin on junctional RhoA signaling appears to involve two pathways:

a) **Tyrosine-dephosphorylation of junctional cortactin** downregulates RhoA signaling by recruiting SRGAP1 (the focus of our manuscript). This is not a feature of

global cortactin KD. Furthermore, SRGAP1 depletion alone increases, rather than decreases, RhoA signaling, consistent with its working as a RhoA antagonist. We further emphasize that this seems to be independent of the capacity for cortactin to regulate junctional F-actin, as the latter was not compromised when cortactin KD cells were reconstituted with the 3YF cortactin mutant.

b) **Depletion of cortactin protein overall** also compromises RhoA signaling. However, this seems to be independent of the SRGAP1 pathway, as it was not ameliorated by SRGAP1 KD. We have not pursued this second pathway, but strongly suspect that it reflects the capacity for cortactin to support the junctional actomyosin cytoskeleton as cortactin KD depletes junctional F-actin and myosin (Han et al., JBC 2014; this manuscript); reconstitution with Arp2/3-uncoupled W22A cortactin depletes junctional actomyosin (this manuscript); and in recent work we have shown that junctional NMII supports RhoA signaling by mechanochemical feedback (Priya et al., NCB, 2015; Priya et al., PLoS Computational Biol, 2017).

2) Both cortactin (heavily) and SRGAP (3) have been implicated in lamellipodial protrusion and integrin-dependent cell migration and it is well established that crosstalk through actomyosin organization/contractility exists between integrin and cadherin adhesions. The authors could counter this argument (at least to a decent extent) by directly testing whether cortactin a (and mutants) and SRGAP1 localize to cell matrix adhesion structures and protrusive cell areas and whether their manipulations affect cell migratory processes (they do actually in figure 8, but simply blame the effects on cadherin adhesion with cannot be a priori concluded).

RESPONSE: We agree that this is an interesting area that was overlooked in the original submission of our paper. Specifically, in Fig 8 we showed that cell movement within confluent monolayers, as well as collective epithelial migration into wounds, was increased by HGF and this effect was reduced by SRGAP1 RNAi.

As these original experiments dealt with cell migration promoted by HGF, we focused especially on this context in our new experiments. We sought to identify basal localization of these proteins in HGF-stimulated monolayer cells. However, as shown in Fig S7b,c, we could not identify co-localization of either SRGAP1 or cortactin with focal adhesions in either control or HGF-treated monolayers, in contrast to the clear increase in junctional SRGAP1 that was observed. Moreover, although cortactin, pYcortactin or SRGAP1 were found in lamellipodia of migrating cells, most especially at the leading margin in scratch wound assays, HGF did not affect the amount of these proteins found there (Fig. S9c-g). We favour the notion that these changes in migration may reflect the impact of change in junctional contractility, consistent with growing evidence (e.g. from the Trepatt group) that cell-cell forces influence collective migration. Of course, it remains possible that changes in junctional RhoA may have had secondary effects on cell-substrate interactions by other pathways. Respectfully, we feel that these are beyond the scope of the present study. However, we have modified the Discussion to make it clear that, in addition to the impact of cell-cell junctions that we favour as an interpretation, we cannot exclude direct or indirect changes to cell-substrate adhesion or locomotility:

“The relationship between junctional relaxation and epithelial motility that we have observed is consistent with increasing evidence that macroscopic patterns of tissue stress and fluidity can influence cell movement within cohesive populations^{34, 49}. Potentially, cortactin and SRGAP1 may also influence epithelial migration by regulating cell-substrate interactions or their dynamics. However, we did not identify either protein at focal adhesions in the epithelial monolayers that we studied, although indirect effects from changes in AJ or junctional signaling cannot be ruled

out.”

3) *Related to this: SRGAP1 is a GAP, but seems to affect RhoA total protein levels rather than specifically RhoA GTP levels at junctions. This is not at all mentioned or discussed and adds to my skepticism towards the models presented. I have not re-read the modeling paper referred to in figure 4 f,g, but here it should definitely and explicitly made clear how GTPase activating function is incorporated and leads to total reduction rather than loss of activation.*

RESPONSE: The reviewer’s point is absolutely true as a general concept. However, in our earlier work we found that steady-state junctional RhoA levels require that it be active. Junctional RhoA was substantially reduced within 30 min after its inhibition with C3-transferase; when its upstream activator, GEF ECT2, was depleted (Ratheesh et al., NCB, 2012); or when it was downregulated by p190B RhoGAP (Priya et al., Nature Cell Biol 2015). Thus, in practice the activity status and steady-state junctional localization of RhoA appear to be closely linked.

Furthermore, because of these experimental observations, the model that we developed was framed in terms of steady-state levels of RhoA signaling and steady-state localization of its known regulators. We apologize that we did not make this sufficiently clear in the current manuscript and have clarified it in the revised description (Modelling section of the Supplemental material).

4) *Technically, the manuscript relies on a lot of quantifications of (Immuno-) Fluorescent images in fixed cells. Not all of these images look of sufficient quality to warrant the current use of this technique which has significant complications because it is difficult to define an internal standard as background and real signal do not fluctuate at the same ratio across experiments or even across slides, or even within slides. For instance in several images in figure 2,3,4,5 and 7 the staining do not look very homogeneous and there is no clear reason think that this is due to a biological reason rather than a technical one. Also, the SRGAP1 levels in junctions in the supplemental figure 3B look much less convincing than those in main figure 3e. The authors could relieve some concern here by performing co-stainings and quantifications with a regular junction marker like E-cadherin itself or b-catenin. Ample suitable antibodies are available.*

RESPONSE: We have endeavoured to improve the quality of the images presented. We agree that the data quality in the submitted PDFs was not as good as it could have been. In particular, one reason for the apparent variability was due to variation in cell height. Our cells do not grow to a uniform height and the apical plane, including the apically-located ZA, can go in and out of focus of a single optical plane between cells that are within a few diameters of one another. One limitation of the previously presented images was that they were taken from single optical planes. Accordingly, we have now:

a) Improved the contrast handling of the images (but with handling applied uniformly to all images presented).

b) Compensated for variation in cell height. Whereas we earlier presented single confocal optical slices, we now present maximum-projection views of the apical-most 3 optical slices to better capture the apical region.

We should emphasize, however, that the quantitative analysis was designed to overcome potential sources of variability.

First, to compensate for the problem of variation in cell height, we worked from image stacks and manually measured fluorescence intensity at the apical junctional region by line scan analysis, with the lines oriented perpendicular to randomly chosen junctions.

This ensured that we were dealing with apical regions that were in the optical plane of analysis. We have now noted this in Methods.

Second, we always studied the junctions between two comparable cells (effectively homotypic cell-cell contacts) – e.g. between KD cells or KD cells that are both reconstituted with transgenes. This was to avoid possible variability associated with heterotypic contacts. Again, this is now noted in Methods.

Third, the data presented are biological replicates (typically from 3 independent experiments). However, each individual biological replicate was the average of at least 25 individual junctions (technical replicates) for most markers that we measured (and at least 60 junctions for GFP-AHPH experiments). Therefore, the final data presented represent at least 75 individual junctions for most markers (and at least 180 junctions for GFP-AHPH). Therefore, we are confident that the quantitative analysis robustly captures the changes in our experiments.

5) I do not understand why the F-actin levels are not restored by SRGAP1 knockdown in HGF-stimulated cells in supplemental figure 6C. If the modeling is correct, how is this explainable?

RESPONSE: We think that a separate pathway regulates F-actin at junctions in response to HGF. In an earlier study (Mangold et al., *Current Biology* 2011, 21, 503-7) we reported that HGF interdicts the anchorage of cortical F-actin to E-cadherin junctions, apparently by disrupting the association of Myosin VI with E-cadherin. Furthermore, this directly or indirectly involved cellular Ca^{2+} signaling. Thus, regulation of junctional F-actin by HGF appears to involve a different signaling pathway to the mechanochemical feedback of Myosin II and RhoA that is captured in our model.

6) In figure 7 I miss the experiments where the effect of cortactin KD (no rescue) on HGF effects and the effect of cortactin YF rescue.

RESPONSE: The aim of these experiments (now in Fig 8) was to test if manipulating cortactin to antagonize the recruitment of SRGAP1 could affect the ability of HGF to downregulate RhoA signaling. Accordingly, we chose to use 3YD cortactin, which we had earlier shown to be capable of reducing junctional SRGAP1 (Fig 3h,i). Therefore, the logical comparison was with cells that expressed WT cortactin under the same conditions (i.e. a transgene which retained the target tyrosine residues that was also controlled for expression levels and any possible transduction effects). As cortactin KD and its reconstitution with 3YF cortactin already reduced junctional RhoA signaling independent of exogenous HGF, these were redundant comparators. Accordingly, we chose the more focused design of the current experiment.

7) In general, in all the KD rescue experiments, the expression controls are lacking, but they are essential to properly judge the experiment. (as KD, lack of KD and rescues may result in the same effects)

RESPONSE: The reviewer makes a very good point. In particular, as the reviewer implies, a key issue is the level to which the transgenes are expressed at adherens junctions, especially relative to the amount of cortactin that is normally found there. In the original submission, we showed that the cortactin transgenes used in this study (WT, 3YF- and 3YD-cortactin) all localize with E-cadherin at adherens junctions to a similar degree (Fig S1d, f) and also by western analysis that these transgenes were expressed to similar extents overall (Fig S1a).

We have now extended this analysis further to test if the transgenes restore

junctional cortactin levels to approach that seen in control cells where cortactin was not depleted. To examine the level of re-expression at the junctions we have now co-stained samples for cortactin (using an antibody that recognizes the transgenes as well as endogenous cortactin), reasoning that this would also allow us to assess the extent to which reconstitution approached the levels of cortactin found in control cells. We found all the cortactin transgenes restore cortactin staining at junctions to the same level as seen for endogenous cortactin in control cells transfected with GFP alone. Note also that our KD approach leads to ~80-90% reduction in total and junctional cortactin levels (Fig S1a,d,e). This data (presented in Fig S1d,e) indicate that in our experiments the cortactin transgenes are expressed at similar levels and restore junctional cortactin to control levels.

Reviewer #3, expert in GTPases

The manuscript by Liang and colleagues describes a potential function of SRGAP on the negative regulation of RhoA activity at junctions. This is an important issue on junction regulation that is not yet fully understood: how dynamic cycles of activation and inactivation of small GTPases are controlled spatially at the molecular level. The authors propose key role of cortactin tyrosine phosphorylation to localize SRGAP at junctions, and thereby modulate RhoA activation levels locally. The authors use a series of quantitative assays to demonstrate the impact on junction contractility, localization of different markers at junctions and the ability of cells to scatter in response to different stimuli. Appropriate controls are shown by using different mutants and rescue experiments. The figures have very nice presentation and are well structured; the text is very clear and readily understood.

Experimentally the work is excellent and the data convincing. However, this reviewer finds conceptual and technical issues that would preclude its publication in its current form. First, conceptually, the distinction between recruitment to junctions and activation has not been properly addressed throughout. This is an important issue as there is clear evidence that many of the RhoA regulators localized at junctions are inactive. Thus recruitment does not always correlate with activation. Second, there are a number of other interpretations to explain the results that were not formally excluded by the authors. These are not tested or discussed in the text. Third, there are concerns with the phospho-mimetic mutant.

Specific points:

1) - *SRGAP1 is a GAP for RhoA, Rac1 and Cdc42 in vitro. In cells and depending on the cellular event, GAPs may have a preference for a particular substrate. However, the authors have not excluded the involvement of Rac1 or Cdc42 in their analysis on junctions or indeed motility. In particular Rac1 can control also contractility, E-cadherin stability at junctions and down-regulate RhoA activation via different mechanisms.*

RESPONSE: This is a very good point. To address this issue we used a location biosensor for GTP-Cdc 42 (incorporating the WASP RBD) and the Rac-Raichu FRET-based activity biosensor and compared their junctional signals in both control and SRGAP1 KD cells. As shown in Fig S4a-d, although GTP-Cdc42 and GTP-Rac are found at apical AJ, their signals were not altered in SRGAP1 KD cells. This contrasts with the increase in markers of RhoA signaling that we found (increased junctional GFP-AHPH, RhoA protein – identified after TCA fixation, and junctional levels of ROCK1 and myosin II). This reinforces our conclusion that SRGAP1 appears to dominantly act as a RhoA antagonist at AJ.

As the reviewer noted, context does influence the impact of SRGAP1. We have also looked for SRGAP1 localization in other regions of the cell, especially cell-substrate interfaces (Fig S7b,c, S9c-g). We did not find any evident co-localization at focal adhesions in control or HGF-treated monolayers; moreover, while we found it at lamellipodia in cells at the migrating front in scratch-wound assays, this was not quantitatively or qualitatively altered by HGF treatment. Accordingly, we have chosen to focus on RhoA regulation at junctions in this manuscript.

2) - *The authors equate recruitment to junctions to activation of a pathway. This is not strictly correct and bypasses important concepts and knowledge of the activation status of signalling molecules at junctions. Interpretation of the data should be revised. For example, if the total level of RhoA protein is increased at junctions and its activation is also increased proportionally (Fig.3i,j), it is difficult to dissect which of these two events is being controlled.*

RESPONSE: The reviewer is absolutely correct. In this revision, we performed an additional experiment that addresses the role of SRGAP1 as a regulator of RhoA activity. We have now measured junctional RhoA signaling when SRGAP1 KD cells are reconstituted with a GAP-deficient mutant as well as wild-type SRGAP1 (Fig 4a-d). As we had shown in the original MS, measures of junctional RhoA signaling (GFP-AHPH, ROCK1, NMIIA and E-cadherin) are increased by SRGAP1 KD. These are restored to control levels when SRGAP1 KD cells are reconstituted with a WT transgene. However, they are not restored by expression of the GAP-deficient mutant. This strengthens our conclusion that SRGAP1 acts as an antagonist of RhoA activity at adherens junctions.

We should note that in our experience the steady-state level of junctional RhoA protein is closely linked to its activation status. Thus, junctional RhoA (identified after TCA-fixation) is reduced within 30 min after its activity is blocked with C3-transferase (Ratheesh et al., Nat Cell Biol 2012). Similarly, depletion of its upstream GEF, Ect2, reduces both the amount of RhoA protein found at junctions as well as its activity, as reported with the Rho-Raichu probe (Ratheesh et al., Nat Cell Biol, 2012) and GFP-AHPH (Priya, NCB, 2015). Accordingly, we have included measures of TCA-fixed RhoA staining as a proxy for changes in RhoA signaling along with other measures that we have shown to require RhoA activity, such as junctional ROCK1 and Myosin IIA levels (Smutny et al., Nat Cell Biol, 2010; Priya et al., Nat Cell Biol 2015). However, we have endeavoured to make it explicit that we are treating these as proxies for changes in steady-state RhoA signaling.

Accordingly, the model that we utilized was developed to analyse steady-state changes in junctional RhoA as well as the upstream regulators that we were studying (Priya et al., Nat Cell Biol 2015). That said, experimental data from our studies (Priya et al., Nat Cell Biol, 2015) as well as what was known from other work (e.g. characterization of Rnd3 and its control of p190B RhoGAP) indicated that when localized to junctions under steady-state conditions these proteins were likely to be predominantly in active (or constitutively-active, for Rnd3) forms.

These conditions of the model, and the experimental evidence for them, were discussed at length in the modeling Supplement to our earlier paper (Priya et al., Nat Cell Biol 2015). Recognizing that we had not provided sufficient introduction in the current manuscript, we have amplified the description of the model so that this is clearer.

3) - The model proposed only uses concentration of a particular molecule, without considering spatial information or activation steps. As presented, the biological impact of such model is reduced. For signalling proteins, the net number of molecules is not relevant, but rather the extent and duration of its activation locally. If inclusion of these additional parameters is not possible in the model, it should be removed or at least these limitations should be raised and discussed in the text.

RESPONSE: The reviewer is absolutely correct. As alluded to in our response to (2), however, the model attempts to analyse a feedback network and its bistable outcomes that we identified at adherens junctions where the steady-state localization of the known proteins is closely linked to their activation status. The assumptions of the model, and the evidence for them, were outlined in detail in our earlier paper, but not adequately summarized in our original manuscript. Accordingly, we have expanded the description of the modeling to provide a better introduction to its conditions and assumptions.

4) - The use of the tyrosine phosphomimetic mutation as aspartic acid on cortactin is not appropriate. Aspartic acid is much smaller than the bulky tyrosine (or phosphotyrosine) residue and Aspartic acid (or glutamic) is normally used to produce phosphomimetic mutants at Serine or Threonine residues. The impact of the YD cortactin mutant binding to

SRGAP1 is not surprising and could be due to structural changes on cortactin protein. Unless more validation is provided, this reviewer is not convinced that it can be interpreted as a phosphomimetic mutant. As this piece of evidence is essential to the conclusions of the manuscript, binding of phosphocortactin to SRGAP should be validated by in vitro phosphorylation or de-phosphorylation of WT cortactin and other known in vivo phospho-cortactin functions should be tested with the YF and YD mutants.

RESPONSE: This is a very thoughtful point. We chose to try to tackle this biochemically, as suggested by the reviewer. To this end, we performed additional experiments to test how its tyrosine-phosphorylation status influences the direct or indirect association of cortactin with SRGAP1 (Fig 3c-e). We expressed GFP-tagged cortactin in HEK293 cells and isolated it by GFP-Trap. Then, to dephosphorylate the isolated GFP-cortactin we incubated it with λ -phosphatase. Alternatively, to increase tyrosine phosphorylation of GFP-cortactin, we co-expressed it with constitutively-active Src^{Y527F}. We confirmed by western analysis (blotting for pY421 cortactin) that the isolated GFP-cortactin was modified as expected (Fig 3c,d). Then we incubated the isolated GFP-cortactin with lysates from Caco-2 cells (which we had treated with cortactin RNAi, to reduce residual cortactin-SRGAP1 complexes). Western analysis of complexes with GFP-cortactin revealed that the associated SRGAP1 was reduced by tyrosine-phosphorylation of cortactin and increased by its tyrosine-dephosphorylation (Fig 3c,e). Thus, these biochemical manipulations reinforce our earlier interpretations inferred from the cortactin mutants. We thank the reviewer for prompting us to test this interpretation more directly.

With regard to the use of the cortactin mutants in our analysis, we would emphasize that the majority of our analyses focus on the 3YF mutant. Only one experiment specifically relies on the 3YD mutant as an analytic tool: that of Fig 8, where we test if manipulating cortactin to antagonize the junctional recruitment of SRGAP1 can affect down-regulation of RhoA by HGF. This builds on our earlier evidence (Fig 7) that SRGAP1 is recruited to junctions by HGF and is necessary for RhoA down-regulation. Indeed, we find that the 3YD mutant does antagonize RhoA down-regulation by HGF, consistent with all the earlier lines of evidence that we present linking cortactin, SRGAP1, RhoA regulation and HGF. We agree, it is formally possible that the impact of the 3YD mutant involves a molecular mechanism independent of tyrosine phosphorylation. Nonetheless, it seems the parsimonious interpretation, especially given our new biochemical evidence. At the very least, it reinforces the notion that cortactin is a key regulator of SRGAP1 localization. Accordingly, we have modified the introduction to this experiment (p 10, bottom paragraph) to read:

“Then, we asked if cortactin mediated the recruitment of SRGAP1 by HGF. As a test of principle, we expressed the 3YD cortactin mutant, which could antagonize SRGAP1 recruitment (Fig. 3h,i), in cortactin KD cells.”

5) - *The data using anti-pY421 cortactin shown in Fig 5 should be interpreted carefully. The anti-phospho antibody will only bind to freely available epitopes (i.e. those not masked by any interacting proteins). If SRGAP1 binds more efficiently to phosphorylated cortactin (or indeed any other associated protein to phospho-cortactin), then one can envisage that there would be less staining with the anti-phospho antibody.*

RESPONSE: This is a judicious note. We now provide two new lines of that further support for the notion that it is desphosphorylation of cortactin that recruits SRGAP1.

a) As outlined above (in response to note #4), the in vitro dephosphorylation of cortactin promotes its binding to SRGAP1.

b) Further, in new experiments we performed E-cadherin-GFP pull-down

experiments from control and HGF-treated cells (Fig S6c). We find that the amount of total cortactin protein is not altered in these E-cadherin-GFP complexes. However, the amount of pY421 cortactin is reduced. Along with the IF data, this supports the notion that junctional cortactin is being tyrosine-dephosphorylated by HGF.

Therefore, while it is difficult, if not impossible, to formally exclude an impact of epitope-masking, in light of these observations we believe that it is parsimonious to hypothesize that it is the dephosphorylation of cortactin that is promoting the interaction with SRGAP1 (rather than the interaction between the two proteins masking pY-cortactin epitopes).

6) - *The localization of active RhoA is very clearly modulated under the different conditions tested. However, the reduction in junctional active RhoA could be due to indirect events. For instance, perturbation of junctional contraction by cortactin perturbations could indirectly modulate the pool of p190RhoGAP as shown in the previous papers from the lab. This possibility was not discussed or eliminated. It would be good to have in the text a broader view of how these two pathways that control RhoA can be coordinated and coexist in the same space.*

RESPONSE: This is a very good point. It is certainly true that the feedback that we are starting to identify at junctions forms the basis for indirect effects (especially once the cells have reached some steady state). We would certainly not wish the reviewer or readers to think otherwise. Accordingly:

a) We have now performed an additional experiment that reinforces the notion that SRGAP1 is acting on junctions through its Rho-inhibitory activity. Here we measured junctional RhoA signaling when SRGAP1 KD cells are reconstituted with either WT SRGAP1 or a mutant lacking the GAP domain (Fig 4a,b). As previously, we find that junctional RhoA is increased in SRGAP1 KD cells, accompanied by an increase in junctional ROCK1, NMIIA and E-cadherin (Fig. 4 c,d,e). This increase does not occur when KD cells are reconstituted with WT SRGAP1, confirming that these effects are specific for manipulation of SRGAP1 expression. However, the increase in RhoA signaling persisted when KD cells were reconstituted with the GAP-deleted mutant. Therefore, the GAP domain and, by implication its known RhoA-inactivating capacity, is necessary for SRGAP1 to regulate junctional RhoA. We believe that this new finding reinforces the conclusion that down-regulation of RhoA plays an important role in allowing SRGAP1 to dampen contractility at cadherin junctions.

b) We have rewritten the Discussion to more explicitly highlight the potential links between the SRGAP1 pathway and the mechanochemical regulation of p190B that we earlier identified. Specifically:

“These findings identify SRGAP1 as a RhoA antagonist at cell-cell junctions, whose capacity to dampen steady-state contractility is coopted by HGF signaling to further relax junctions. They further imply that SRGAP1 may function in two regimes that depend on its levels at junctions. Interestingly, in earlier work we also identified p190B RhoGAP as a potent antagonist of RhoA signaling at the ZA¹⁵. However, whereas SRGAP1 is found at steady-state zonulae adherente, p190B was excluded by mechanochemical feedback from stabilized cortical NMII²⁸, which anchored ROCK1 at the ZA to antagonize Rnd3, the mechanism responsible for recruiting p190B to the cortex²⁸. This feedback network generates bistable outcomes for RhoA signaling²⁸ and can also define the size and borders of the RhoA signaling zone⁴⁷. However, once NMII-RhoA feedback is established at the mature ZA, its capacity to

exclude p190B implied that another GAP might be required to modulate RhoA signaling.

From this perspective, SRGAP1 can be understood as a RhoA antagonist that can act orthogonal to the NMIIA-RhoA network. Indeed, incorporation of SRGAP1 into a computational model of the NMIIA-RhoA network demonstrated that low levels of SRGAP1 could modulate RhoA signaling within a persistent range of activity, but above some threshold, SRGAP1 then effectively drove the RhoA zone into a relatively inactive state. We suggest that the behaviour of steady-state monolayers may correspond to the first regime, where GTP-RhoA signaling zone is maintained despite the presence of SRGAP1, whereas HGF acts to drive the system into the second regime. The precise relationship between SRGAP1 and p190B will be an interesting issue for future research. One possibility is that the threshold for SRGAP1 to enter its second regime may reflect when sufficient SRGAP1 has accumulated to overcome buffering by the NMII-RhoA feedback network.”

c) Finally, we would emphasize that our current data indicate cortactin influences junctional RhoA signaling by two pathways:

i) **Tyrosine-dephosphorylation of junctional cortactin** downregulates RhoA signaling by recruiting SRGAP1 (the focus of our manuscript). This is not a feature of global cortactin KD. Furthermore, SRGAP1 depletion alone increases, rather than decreases, RhoA signaling, consistent with its working as a RhoA antagonist. We further emphasize that this seems to be independent of the capacity for cortactin to regulate junctional F-actin, as the latter was not compromised when cortactin KD cells were reconstituted with the 3YF cortactin mutant.

ii) **Depletion of cortactin protein overall** also compromises RhoA signaling. However, this seems to be independent of the SRGAP1 pathway, as it was not ameliorated by SRGAP1 KD. We strongly suspect that this pathway reflects the capacity for cortactin to support the junctional actomyosin cytoskeleton as cortactin KD depletes junctional F-actin and myosin (Han et al., JBC 2014; this manuscript); reconstitution with Arp2/3-uncoupled W22A cortactin depletes junctional actomyosin (this manuscript); and in recent work we have shown that junctional NMII supports RhoA signaling by mechanochemical feedback (Priya et al., NCB, 2015; Priya et al., PLoS Computational Biol, 2017).

We have not explored the second pathway in any depth, as we felt that it ran the risk of confusing the reader (it certainly confused us!) and we also think that this is better left for a separate study. However, we do discuss this in the final paragraph of the Discussion. We agree with the reviewer that this highlights the likely complexities of the cytoskeleton and RhoA signaling at the ZA.

“It should be noted that junctional RhoA signaling was also compromised by cortactin KD alone. However, this effect was not altered by SRGAP1 KD, suggesting that it reflects another pathway for cortactin to influence RhoA. We suggest that this SRGAP1-independent pathway is likely to reflect the ability of cortactin to support actin assembly^{17, 18}, which helps recruit NMII to junctions^{12, 14}. Indeed, we found that W22A cortactin, which cannot support Arp 2/3, reduced junctional NMII and tension. As NMII is required to support junctional RhoA signaling²⁸, its loss in cortactin KD cells could compromise RhoA. Cortactin may then have two contributions to RhoA signaling at the ZA: a basal requirement that reflects its vital contribution to biogenesis of junctional actomyosin, and a regulatory role where its degree of tyrosine de/phosphorylation tunes RhoA signaling through recruitment of SRGAP1. This highlights the complex inter-relationship between actomyosin biogenesis and cell signaling at adherens junctions.”

7) - *The use of cell motility to confirm a potential effect of SRGAP1 on junction relaxation is problematic. It has not been shown in the cellular system presented whether SRGAP1 interferes with focal adhesion turnover or motility per se. Without these controls it is difficult to conclude that the motility defects are caused to junction stability and contractility.*

RESPONSE: This is a very reasonable point. In an effort to pursue this we have tested whether SRGAP1 may localize to cell-substrate interfaces in control cells or cells stimulated with HGF. To date, we have been unable to find any such localization, either with indirect immunofluorescence for the endogenous protein or with an expressed transgene. We do identify cortactin, pY-cortactin and SRGAP1 at the lamellipodia of migrating cells, principally at the free margin of cells in scratch-wound assays, but the levels of these proteins are not altered by HGF. Thus, we still favour a role for tension at cell-cell junctions, something that is consistent with increasing evidence (e.g. from the Trepap group) showing that cellular stresses influence collective migration. It is certainly possible that there is a pool below the detection limits of our current assays or that there are indirect effects on cell-substrate adhesion that are responding to the change in junctional RhoA signaling and contractility. Respectfully, we feel that pursuing these possibilities warrants a project in itself and is beyond the scope of this current, rather full, manuscript. We have, however, revised the Discussion to incorporate these possibilities:

“The relationship between junctional relaxation and epithelial motility that we have observed is consistent with increasing evidence that macroscopic patterns of tissue stress and fluidity can influence cell movement within cohesive populations^{34, 49}. Potentially, cortactin and SRGAP1 may also influence epithelial migration by regulating cell-substrate interactions or their dynamics. However, we did not identify either protein at focal adhesions in the epithelial monolayers that we studied, although indirect effects from changes in AJ or junctional signaling cannot be ruled out.”

Reviewer #4, expert in cortactin

In this manuscript by Liang and Yap, the authors present evidence that cortactin dephosphorylation recruits the RhoA antagonist srGAP1 to epithelial zonula adherens junctions to downregulate contractility. The authors demonstrate that nonphosphorylatable cortactin (3YF) decreases contractility and RhoA activity at cell junctions by preferentially recruiting srGAP1. Hepatocyte growth factor stimulates cell motility by decreasing cortactin phosphorylation and recruiting srGAP1 to decrease contractility.

This work represents a potential advance in understanding the complementary but distinct roles of cortactin in actomyosin biogenesis and RhoA regulation at cell junctions. However, in some places the data lacks a degree of experimental rigor necessary to support the conclusions drawn by the authors. Thus, without significant revisions it does not currently meet the standards of Nature Communications.

Major Points:

1) The expression levels of cortactin mutants in knockdown cells (shown in Fig S1a, b) is not at the level of endogenous protein. The mCherry-fusions expressed in some cases are below the level of residual endogenous cortactin after knockdown. Given the dependence of the conclusions in Figures 1, 2, 3, 4, and 7 on the expression of cortactin mutants, experiments may need to be repeated with more robust controls for mutant expression relative to wild-type levels (and in excess of residual endogenous protein).

RESPONSE: As the reviewer implies, a key issue is the level to which the transgenes are expressed at adherens junctions, especially relative to the amount of cortactin that is normally found there. In the original submission we showed that WT cortactin and the phosphomutants localize to junctions to a comparable degree (data now found in Fig S1d,f).

To examine the level of re-expression at the junctions we have now co-stained samples for cortactin (using an antibody that recognizes the transgenes as well as endogenous cortactin). We find that all the cortactin transgenes restore cortactin staining at junctions to the same level as seen for endogenous cortactin in control cells transfected with GFP alone. These data (presented in Fig S1d,e) indicate that in our experiments the cortactin transgenes are expressed at similar levels and restore junctional cortactin to levels comparable to those displayed by endogenous cortactin in control cells.

2) Methods for quantification of E-cadherin, RhoA, ROCK1, srGAP1, cortactin, and pY-cortactin are dependent on a peak intensity value based on a line trace across a used-defined junction on cells. While Western blotting confirms expression levels of their proteins are unchanged during perturbations, the method for quantification of GFP-AHPH, using the ratio of mean junction fluorescence relative to mean cytoplasmic fluorescence, would be more rigorous in quantifying the enrichment of these proteins at junctions.

RESPONSE: Formally, this is certainly possible. To test whether this affects our results in practice, we performed a test quantification using correction for cytoplasmic fluorescence. We show these data below, with each panel labeled for its equivalent in the Figures (where we quantitate using our earlier technique). As is evident, although the precise values change, of course, the alternative quantitation method does not change the patterns. As this alternative method doesn't affect the interpretation of the data, we believe that our original system, based on the peak values of fluorescence intensity is reliable. Therefore we respectfully prefer to continue with it.

Quantification of junctional E-cadherin, RhoA, ROCK1, cortactin, pY421-cortactin and SRGAP1 normalized to cytoplasmic levels of the protein from key experiments of the study.

3) What are the details of the interaction screen that reveals srGAP1 as an interaction partner of cortactin-3YF? The authors state that this interaction could be direct or indirect, a hypothesis that is testable with recombinant proteins. Recombinant proteins would also allow for testing dephosphorylated cortactin (incubated with a phosphatase) as opposed to nonphosphorylatable cortactin-3YF.

RESPONSE: We have now included a description of the interaction screen in the Supplementary Methods section.

With due respect, we feel that a test of in vitro binding with recombinant proteins is beyond the scope of this revision. As noted below (in response to point #4) our pilot studies suggest that multiple domains of SRGAP1 contribute to its junctional localization, implying that the biochemical basis for recruitment may be multifactorial. Further, both cortactin and SRGAP1 are large proteins and would require substantial effort to optimize in vitro production.

However, we have performed an additional experiment to test more directly whether the tyrosine-phosphorylation status of cortactin can influence its capacity to bind SRGAP1 (Fig 3c-e). To this end, we expressed GFP-tagged cortactin in HEK293 cells and isolated it by GFP-Trap. Then, to dephosphorylate the isolated GFP-cortactin we incubated it with λ -phosphatase. Alternatively, to increase tyrosine phosphorylation of GFP-cortactin, we co-expressed it with constitutively-active Src^{Y527F}. We confirmed by western analysis (blotting for pY421 cortactin) that the isolated GFP-cortactin was modified as expected (Fig 3c,d). Then we incubated the isolated GFP-cortactin with lysates from Caco-2 cells (which we had treated with cortactin RNAi, to reduce residual cortactin-SRGAP1 complexes). Western analysis of complexes with GFP-cortactin revealed that the associated SRGAP1 was reduced by tyrosine-phosphorylation of cortactin and increased by its tyrosine-

dephosphorylation (Fig 3c,e). Thus, these biochemical manipulations reinforce our earlier interpretations inferred from the cortactin mutants. We thank the reviewer for prompting us to test this interpretation more directly.

4) *IF the interaction with srGAP1 is direct, the binding site on cortactin is contained within what is likely to be an extended unstructured region. Which regions of srGAP1 bind to cortactin? Do finer point mutations in srGAP1 that disrupt cortactin binding affect srGAP1 junctional recruitment, Rho activity, and junctional recoil?*

RESPONSE: As a first step to addressing this question, we reconstituted SRGAP1 KD cells with mutants deleted in known domains (F-Bar, SH3 and GAP). We have found that all localize to junctions, which is consistent with Ronen Zaidel-Bar's earlier observation that multiple domains are necessary for *C. elegans* SRGP-1 to localize to junctions (JCB 2010, 191, 76-769). We illustrate that data here but prefer to omit it from the revised manuscript as it doesn't lead to a conclusive outcome.

With respect, and in light of this localization data, we feel that these are substantive (and challenging) questions that deserve a dedicated study of their own.

(A) Representative images of SRGAP1 mutants at the apical region of Caco-2 cells. (B) Quantification of junctional SRGAP1 mutants normalized to cytoplasmic levels.

5) *What degree of knockdown is achieved with SRGAP siRNA? Can the blot in Figure S3A or the images in S3B be quantified? While the use of two different siRNAs is indicative of specificity, rescuing the SRGAP knockdown cells with a siRNA-resistant mutant is the gold standard in confirming specificity, as the authors have done for their cortactin knockdown experiments.*

RESPONSE: We see ~ 70-80% depletion of junctional SRGAP with siRNA. These data are now quantified and presented in Fig S4g.

In further experiments, we have performed the gold-standard experiment as suggested by the reviewer (Fig 4a-e). Here we reconstituted SRGAP1 KD cells with WT SRGAP1 and also a transgene lacking the GAP domain. As described in the original manuscript, we found that RhoA signaling was increased in SRGAP1 KD cells (as measured with GFP-AHPH and also proxies that include junctional ROCK1, Myosin II and E-cadherin). This increase was blocked if KD cells were reconstituted with WT SRGAP1, but not by the GAP-deficient mutant, even though it localized to junctions as did WT SRGAP1. Therefore, this confirms the specificity of the KD effect and also strengthens the notion that downregulation of RhoA activity is key to the impact of SRGAP1 on AJ. We thank the reviewer for this suggestion.

6) The authors refer to cortactin dephosphorylation throughout the manuscript, making inferences using phosphomimetic or nonphosphorylatable mutants. As they mention in their discussion, screening for potential phosphatases that regulate cortactin's effects in cell junctions would strengthen this argument considerably and demonstrate that the phenomena observed are due to enzymatic dephosphorylation that occurs in physiological conditions. What is known about the mechanism of dephosphorylation in response to HGF treatment?

RESPONSE: Respectfully, we feel that this is a problem that is beyond the scope of what is already a rather long and full paper (as indicated by the Editor). A number of protein tyrosine phosphatases have been reported to be downstream of HGF signaling, including Shp-2 (e.g. Schaeper JCB 2000, 149, 1419-32). However, we feel that this question warrants a systematic analysis of its own.

7) *It would seem important for the authors to confirm that their pY421 antibody is phospho-specific? This could be done testing for signal in cortactin-3YF cell lines using both total cortactin and phospho-specific antibodies. The concern is that residual cortactin immunoreactivity could be contributing to the signal.*

RESPONSE: We thank the reviewer for this suggestion. We have tested this in two ways. 1) As suggested by the reviewer, we used the pY421 Ab to immunostain cortactin KD cells reconstituted with the phosphomutant transgenes (Fig S1g,h). We found that junctional pY421 staining was abolished by cortactin KD, reconstituted with WT cortactin, reduced in cells expressing cortactin-3YF and somewhat elevated in those expressing cortactin 3YD.

2) Additionally, in new experiments (Fig 3c-e) we manipulated the tyrosine phosphorylation of cortactin-GFP either by co-expressing it with constitutively-active Src^{527F} in HEK293 cells (to promote tyrosine-phosphorylation) or by in vitro dephosphorylating isolated cortactin-GFP by incubation with λ -phosphatase. In Western analysis, we found that pY421 reactivity was decreased by λ -phosphatase and increased by co-expression with Src^{527F}.

Overall, we feel that these provide reasonable evidence that our pY421 Ab is phospho-specific.

8) *Figure 8A and Supplementary Movie 1 have a significant confounding factor in the amount of cell division occurring in the HGF conditions. Some of the longest motility tracks seen in the HGF condition are in the case of a cell that divides and rapidly splits. This data should be quantified with those cells expressly excluded.*

RESPONSE: The reviewer is absolutely correct. In fact, we had manually excluded those cells in the quantitative analysis shown in Fig 8 (now Fig 9). We apologize for omitting to state this: it has now been corrected in the Supplemental Methods.

9) *In their conclusion, the authors suggest that srGAP1 is present at cell junctions at a low concentration range that their model indicates will modulate RhoA activity without crossing the threshold to reach an inactive RhoA state. Can the authors, by use of quantitative immunofluorescence in conjunction with immunoblotting, estimate the relative concentrations of srGAP1, cortactin, and RhoA at cell junctions to validate this model?*

RESPONSE: This is a very interesting suggestion. However, a robust experimental test of

the model requires measurement of the absolute number of molecules present at junctions, rather than estimates of relative differences in concentrations by immunofluorescence (which is itself a difficult task due to the different affinities of the various antibodies that are used). Therefore, we hope that the reviewer and editor agree that while a more quantitative validation of the model is an interesting suggestion, it warrants a study of its own.

10) *The authors suggest that the W22A mutant of cortactin inhibits cortactin interaction with the Arp2/3 complex. It is possible that cortactin can indirectly recruit and activate the Arp2/3 complex in cells, perhaps through Nck1:N-WASp binding to the cortactin proline-rich region. It would seem imperative for the authors to quantify Arp2/3 localization to cell junctions in each of their cortactin rescue cell lines?*

RESPONSE: This is a good point and we have performed the suggested experiments (Fig S2c,d). Junctional Arp3 levels are reduced by cortactin KD. However, they are restored by both 3YF-cortactin and 3YD cortactin to similar levels as WT cortactin. Thus, it seems unlikely that the impact of these phospho-mutants involves changes in Arp2/3 localization, something that is also supported by our observation that these mutants do not alter steady-state F-actin levels.

Minor Points:

1. The authors suggest that cortactin is orthogonal to their RhoA regulatory network outlined in Figure 4F. Have they tested for cortactin knockdown effects on p190B or Rnd3 expression and localization? This may be especially relevant in the context of Binamé et al, JBC 2016 showing an interaction between cortactin and p190A.

RESPONSE: It is worth noting here that in our manuscript we identify two pathways for cortactin to influence junctional RhoA signaling.

a) ***Tyrosine-dephosphorylation of junctional cortactin*** downregulates RhoA signaling by recruiting SRGAP1 (the focus of our manuscript). This is not a feature of global cortactin KD. Furthermore, SRGAP1 depletion alone increases, rather than decreases, RhoA signaling, consistent with its working as a RhoA antagonist. We further emphasize that this seems to be independent of the capacity for cortactin to regulate junctional F-actin, as the latter was not compromised when cortactin KD cells were reconstituted with the 3YF cortactin mutant.

b) ***Depletion of cortactin protein overall*** also compromises RhoA signaling. However, this seems to be independent of the SRGAP1 pathway, as it was not ameliorated by SRGAP1 KD. We strongly suspect that it reflects the capacity for cortactin to support the junctional actomyosin cytoskeleton as cortactin KD depletes junctional F-actin and myosin (Han et al., JBC 2014; this manuscript); reconstitution with Arp2/3-uncoupled W22A cortactin depletes junctional actomyosin (this manuscript); and in recent work we have shown that junctional NMII supports RhoA signaling by mechanochemical feedback (Priya et al., NCB, 2015; Priya et al., PLoS Computational Biol, 2017). However, we have not pursued this pathway, because we feel that including it would confuse the reader and distract from the focus of this manuscript (it certainly confuses us!). However, we do discuss this in the final paragraph of the Discussion:

“It should be noted that junctional RhoA signaling was also compromised by cortactin KD alone. However, this effect was not altered by SRGAP1 KD, suggesting that it reflects another pathway for cortactin to influence RhoA. We suggest that this SRGAP1-independent pathway is likely to reflect the ability of cortactin to support

actin assembly^{17, 18}, which helps recruit NMII to junctions^{12, 14}. Indeed, we found that W22A cortactin, which cannot support Arp 2/3, reduced junctional NMII and tension. As NMII is required to support junctional RhoA signaling²⁸, its loss in cortactin KD cells could compromise RhoA. Cortactin may then have two contributions to RhoA signaling at the ZA: a basal requirement that reflects its vital contribution to biogenesis of junctional actomyosin, and a regulatory role where its degree of tyrosine de/phosphorylation tunes RhoA signaling through recruitment of SRGAP1. This highlights the complex inter-relationship between actomyosin biogenesis and cell signaling at adherens junctions.”

2. Two typos are present:

- a. Downregulation is one word in the manuscript title but two words in the title of Figure 5.
- b. The symbol for viscosity coefficient is mu in the equations but eta in the text of the Supplemental Section “Two-photon laser ablation and tension measurements”

RESPONSE: Corrected. Thank you!

Reviewers' comments:

Reviewer #1 (Remarks to the Author):

In my opinion, the authors were largely unresponsive to the previous critique. It was requested that they provide the cortactin staining for the images in Figure 1c. These were not included. Similarly, in my previous review, I had concern with the fact that the authors were attributing all their effects to cortactin's effects on cell-cell adhesion without having ever examined cell-matrix events. Instead of examining cell-matrix adhesion in the cell lines expressing the mutant cortactins, they state they could not identify co-localization of either SRGAP1 or cortactin to focal adhesions. This response is inadequate in my opinion because: 1) it is trivial to perform an adhesion assay and to examine the phenotype of cells plated on matrix proteins, and 2) the fact that the authors cannot visualize cortactin in focal adhesions is not evidence for the lack of an effect on cell-matrix adhesion. Similarly the authors did not perform the requested biochemical measures of RhoA-GTP loading or E-cadherin levels. Here again, I would argue that numerous laboratories are using these assays to corroborate immunofluorescence findings. For the sake of comparing this work to the other studies in the field, the RhoA-GTP loading assays should be performed. Finally, I asked the authors to explore the proposed interaction between dephosphorylated cortactin and SRGAP1. They have done some preliminary experiments which they included in the letter, but have not added to the manuscript. Without this data, I believe the manuscript is largely speculative and too preliminary. I do not believe the manuscript was adequately revised. Hence, many of the initial problems still persist.

Reviewer #2 (Remarks to the Author):

I am mostly satisfied and convinced by the controls and discussions added to the revised study by the others, as well as by their rebuttal and explanations to some of my comments. I find this manuscript suitable and acceptable for publication in NCOMM now.

Reviewer #3 (Remarks to the Author):

The authors did a very good job to address the questions raised by the reviewers. The controls added are convincing and new explanations improve the clarity of the text.

The key point that was not addressed was to demonstrate that the junctional RhoA equates to active RhoA – as it stands, the conclusion is an over-interpretation. This is an important point as it goes against standard concepts in the GTPase signalling field. In other words, what is needed is to show which proportion of active RhoA at junctions (by biosensors) is present with respect to the localized total RhoA protein in the same location (antibody or expression tag). This can considerably strengthen their thesis. The argumentation presented in the rebuttal (page 7 and 11) does not address the point raised: (i) they just argue that in past publications the use of C3 toxin, depletion of Ect2 or expression of p190RhGAP leads to reduction of active RhoA levels; i.e. RhoA responded as predicted by manipulation of its regulators and that the biosensor(s) detected active RhoA; (ii) steady state levels can also be expressed as the ratio active/total pool; (iii) the model proposed does

not consider/include the premise of how active pool versus total pool localized at junctions changes upon different stimuli; and (iv) the quantification correction in fig 2 (junctional over cytoplasmic levels of the biosensor) assesses enrichment of the biosensor at cell-cell contacts. Furthermore, the importance of controlling for specific activity can be found in the revised version, where the authors successfully demonstrated the amount of pY-cortactin corrected by levels of different cortactin constructs found at cell-cell contacts (Suppl Figure 1 f, h). Similar controls should be done for RhoA.

Other points:

1. The use of SRGAP1 mutant depleted of the GAP domain is not the appropriate mutant to demonstrate GAP-dependence. Technically, full-length SRGAP mutated at the key catalytic residue in the GAP domain is commonly used. The GAP-deleted mutant is considered a dominant-negative construct as shown for other GAPs, and most likely works by preventing the localization of the endogenous SRGAP to junctions. Experiments presented in the rebuttal letter showing the localization of different domains of SRGAP1 at junctions confirm this possibility. Thus the new experiments confirm the requirement of SRGAP and its correct positioning at contacts to modulate RhoA. The interpretation of the results should be modified to account for this. In addition, the term "GAP-deficient" means catalytically inactive (i.e. point mutation to abolish catalysis). For clarity, "GAP-deficient" used in the main text should be changed to "GAP-depleted" to reflect the dominant negative nature of the construct used.

2. The new intensity quantification of proteins at junctions using the three uppermost confocal sections (rather than a single confocal section) is an improvement against variability across samples and treatments. The authors should clarify which experiments were re-quantified using this method or if the new analysis was applied across all results.

Reviewer #4 (Remarks to the Author):

The efforts by Liang and Yap to respond to reviews of their manuscript are greatly appreciated. Especially informative changes include coimmunoprecipitation experiments in Figure 3 showing a biochemical interaction between cortactin and SRGAP1 that is dependent on cortactin phosphorylation state. Figure 4 is also strengthened by rescuing RhoA inhibition in SRGAP knockdown cells with both WT and GAP-deficient SRGAP1.

However, without addressing relative expression levels of proteins in cortactin knockdown experiments as well as improving image quality in a select number of immunofluorescence experiments, this paper cannot be recommended for acceptance.

Major Points:

1. While the efforts of the authors to control for cortactin transgene localization to adherens junctions (Fig. S1d,e) are noted, immunoblots in Figures S1a and S1b still show that transgenes are expressed at much lower levels of endogenous protein. In Figure S1a transgenes are expressed at much lower levels than cortactin in the control cells, and at levels comparable to the remaining endogenous protein after knockdown. In Figure S1b the transgenes are expressed at lower levels

that the remaining endogenous cortactin. Experiments performed with this degree of knockdown and reexpression cannot sufficiently isolate the effects of the transgene from that of endogenous cortactin. In fact, data in Figure S1d showing cortactin localization to junctions in these cells could represent a combination of endogenous and transgenic protein based on immunoblots.

2. As mentioned by Reviewer 1, image quality needs to be improved, especially with regards to the pY421 antibody used for immunofluorescence. Images in Figures S1c, S1g, 6l, and S9b are examples of anti-pY421 blots that have high background or a speckled appearance that make it difficult to interpret quantification based on peak intensity in the adherens junction at points identified by the user. Interpretations of these images could be strengthened by overlaying another marker of the junction in a merged image, showing where line traces were performed, or comparing cytoplasmic to junction signal as the authors demonstrated in their rebuttal. Anti-Arp2/3 images in Figure S2C could benefit from the same methods.

Minor Points:

1. It is agreed that identifying the relevant cortactin phosphatase may be outside the scope of this work. However, authors should take care to not refer to “dephosphorylated cortactin” in their text unless supported by the experiment. Work done in Figure 3C using lambda phosphatase shows evidence of the effects of dephosphorylation on cortactin-SRGAP1 interaction. However, experiments performed using a 3YF mutant observe behavior of nonphosphorylatable cortactin, and do not directly speak to dephosphorylation events regulating Rho signaling.
2. Efforts to perform experiments in Figures 3c-e are much appreciated and increase the rigor of this work significantly. A better representative immunoblot in Figure 3c could be chosen to reflect a decrease in pY421 upon lambda phosphatase treatment shown in the quantification in Figure 3d.
3. The added detail in the Methods section with regards to the interaction screen is very informative. However, if scores of interactions of phosphorylated cortactin with SRGAP1 as well as other top hits were included this would be both informative and impactful to the field. In the body of the text, mentioning the screen is mass spectrometry-based may also be helpful to readers.
4. Experimental details for Movies 1 and 2, including plating, labeling, and imaging conditions should be added to the Methods section.

RESPONSE TO REVIEWERS

General:

We thank the reviewers for their thoughtful feedback. In this revision we have performed additional experiments and further developed the text in response to the comments of Reviewers 1,3 and 4, guided by feedback from the editor.

Reviewer #1:

1) In my opinion, the authors were largely unresponsive to the previous critique. It was requested that they provide the cortactin staining for the images in Figure 1c. These were not included.

Response: Our apologies: this is now included in Supplemental Fig. 2. What we now show are the corresponding images for cortactin and the relevant transgene (i.e. defining the cells and junctions that are marked with asterisks and arrows, respectively, in Fig 1C). We do this separately for the E-cadherin, F-actin and NMIIA images. Because of the unwieldy size of the images, we have included them all in the Supplementary figure and retained the asterisks and arrows in the main figure.

2) Similarly, in my previous review, I had concern with the fact that the authors were attributing all their effects to cortactin's effects on cell-cell adhesion without having ever examined cell-matrix events. Instead of examining cell-matrix adhesion in the cell lines expressing the mutant cortactins, they state they could not identify co-localization of either SRGAP1 or cortactin to focal adhesions. This response is inadequate in my opinion because: 1) it is trivial to perform an adhesion assay and to examine the phenotype of cells plated on matrix proteins, and 2) the fact that the authors cannot visualize cortactin in focal adhesions is not evidence for the lack of an effect on cell-matrix adhesion.

Response: The reviewer is right and we have now performed the assays. We tested cell-matrix (fibronectin) adhesion in two contexts: in characterizing the potential effects of cortactin KD and its reconstitution with cortactin mutants; and in testing the impact of HGF and how this may involve SRGAP1. What we find is that:

a) *Cortactin KD and reconstitution with cortactin mutants* (included in Supplementary Fig. S3c). We find that cell-matrix adhesion is reduced in cortactin KD cells and restored by reconstitution with WT cortactin. As the reviewer indicated, this demonstrates an impact of cortactin on cell-matrix adhesion even though it was not evident at focal adhesions. Cell-matrix adhesion was not restored by the Arp2/3-uncoupled W22A mutant, suggesting that this effect reflects regulation of actin assembly by cortactin.

However, cell-matrix adhesion was restored as effectively by the cortactin 3YF and 3YD mutants as it was by WT cortactin. These findings contrast clearly with the impact that reconstituting these mutants has on cell-cell junctions, where 3YF cortactin failed to restore myosin IIA, contractile tension and RhoA signaling, despite restoring steady-state F-actin levels. Thus, the impact of the tyrosine non-phosphorylated cortactin mutant maps *relatively* selectively to cell-cell junctions rather than to cell-substrate adhesions. We thank the reviewer for this suggestion, as it reinforces the motivation for better understanding how tyrosine non-phosphorylated cortactin affects junctional tension and contractility.

b) *HGF and the impact of SRGAP1* (data in Supplementary Fig. S10h). Here we found no detectable change in cell-matrix adhesion when cells were treated with HGF, nor was this affected by SRGAP1 depletion.

We would note, however, that we do not want to give the impression that we are categorically

excluding potential effects on cell-substrate interactions. In the previous revision we wrote in the Discussion that:

“Potentially, cortactin and SRGAP1 may also influence epithelial migration by regulating cell-substrate interactions or their dynamics. However, we did not identify either protein at focal adhesions in the epithelial monolayers that we studied, although indirect effects from changes in AJ or junctional signaling cannot be ruled out.”

We have expanded this further in light of the new data:

“Potentially, cortactin and SRGAP1 may also influence epithelial migration by regulating cell-substrate interactions or their dynamics. Indeed, although we did not identify it at focal adhesions, cortactin KD reduced cell-matrix adhesion in an apparently Arp2/3-dependent manner. However, cell-matrix adhesion was restored by cortactin phospho-mutants as effectively as it was by WT cortactin; nor did HGF or SRGAP1 KD affect cell-matrix adhesion. Thus changes in junctional contractility associated with tyrosine non-phosphorylation cortactin correlated better with altered migration than did change in cell-matrix adhesion. The notion that changes in ZA contractility may influence epithelial migration is consistent with increasing evidence that macroscopic patterns of tissue stress and fluidity can affect cell movement within cohesive populations^{37, 51}.”

3) Similarly the authors did not perform the requested biochemical measures of RhoA-GTP loading or E-cadherin levels. Here again, I would argue that numerous laboratories are using these assays to corroborate immunofluorescence findings. For the sake of comparing this work to the other studies in the field, the RhoA-GTP loading assays should be performed.

Response:

1) Rho: We have endeavoured to perform GTP-RhoA pull-down assays on multiple occasions in cortactin RNAi cells, without meaningful results. As can be seen from the example (shown below for the reviewer), although the GTP-RhoA and GDP-RhoA controls perform as expected, the total GTP-RhoA levels were not altered in cortactin KD cells compared with control cells. Nonetheless, these are the exact conditions where we have clear evidence that RhoA signaling and contractility at the ZA are compromised (measured using location and activity biosensors for GTP-RhoA, as well as a variety of downstream proxies for the pathway). One likely explanation for this discrepancy is that the pool of GTP-RhoA at the ZA is a relatively small one compared with the total cellular pool that is being sampled in the pull-down assays. (The apparent prominence of the ZA pool when assayed with a location biosensor such as GFP-AHPH may then reflect its local concentration.) This would be consistent with what is seen in other circumstances where RhoA signaling is spatially confined, as has motivated the search for markers for the spatio-temporal distribution of GTP-RhoA. Accordingly, since we could detect no change with cortactin KD, we have not endeavoured to test this further with the transgene-reconstituted cell lines.

As an alternative, we used a FRET-based activity sensor to measure RhoA activation at the ZA. This showed that cortactin KD reduces RhoA activation, consistent with what we had concluded from the other assays. This data is now included in Supplementary Fig 3f,g. We should note that it was not possible to perform this experiment with the reconstituted cortactin transgenes, as their GFP tag overlapped with the fluorophores used in the FRET sensor. We feel that this corroborates what we had observed with the GFP-AHPH location biosensor, and the latter has the additional value of monitoring endogenous GTP-RhoA.

2) Regarding E-cadherin: we were unclear as to what the reviewer was asking us to measure. We had already measured cellular E-cadherin levels with the principal cortactin mutants (Fig S1a,b) where we measured both total cellular levels of E-cadherin by western analysis as well as measured surface expression of the protein by surface trypsinization. We presumed that the reviewer was asking for a biochemical measure of E-cadherin (or changes in E-cadherin) at the ZA, specifically. However, it is not clear that there is a reliable assay for this pool, as it is a **subpopulation** of the cadherin present at the cell-cell contacts. As we have shown (Wu et al., NCB, 2014), there is extensive E-cadherin throughout the cell-cell contacts between epithelial cells, both at the ZA and the lateral contact surfaces found below the ZA. We do not know of any reliable biochemical way to separate these two pools. Assays for detergent solubility have been used in the older literature as indices of cytoskeletal association. However, we have shown that both the ZA and the lateral pools of cadherin are physically linked to actomyosin: in the case of the lateral pool, actomyosin actually causes planar movement of cadherin clusters (Wu et al., NCB 2014). So it was not clear to us whether detergent-solubility would be effective to discriminate the ZA pool. Moreover, although detergent-solubility assays were popular in the past (and have persisted in some of the literature), they are not necessarily diagnostic for cytoskeletal association, since they have also been used as assays for segregation in lipid rafts.

Therefore to provide an independent assay for change in the ZA pool of cadherin, we used FRAP assays to measure E-cadherin stability (now included in Fig. 1 g,h). Indeed, we find that the immobile fraction of E-cadherin-GFP is reduced by cortactin depletion, consistent with the reduced steady-state levels that we earlier observed. Further, this is restored by WT, but not 3YF cortactin. We consider that these findings support our earlier results, implying that the 3YF mutant impairs the ability of cells to stabilize E-cadherin to form the ZA. Of note, we earlier found that this stabilization of E-cadherin requires both myosin and Rho (Smutny et al., NCB 2010; Ratheesh et al., NCB 2012), consistent with our present demonstration that the 3YF mutant impairs both of these elements at the ZA.

4) Finally, I asked the authors to explore the proposed interaction between dephosphorylated cortactin and SRGAP1. They have done some preliminary experiments which they included in the letter, but have not added to the manuscript.

Response: At the Editor's suggestion, we have now included the data shown in the response letter as Supplementary Fig. S4 c-e.

Reviewer #3:

The authors did a very good job to address the questions raised by the reviewers. The controls added are convincing and new explanations improve the clarity of the text.

1) The key point that was not addressed was to demonstrate that the junctional RhoA equates to active RhoA – as it stands, the conclusion is an over-interpretation. This is an important point as it goes against standard concepts in the GTPase signalling field. In other words, what is needed is to show which proportion of active RhoA at junctions (by biosensors) is present with respect to the localized total RhoA protein in the same location (antibody or expression tag). This can considerably strengthen their thesis. The argumentation presented in the rebuttal (page 7 and 11) does not address the point raised: (i) they just argue that in past publications the use of C3 toxin, depletion of Ect2 or expression of p190RhGAP leads to reduction of active RhoA levels; i.e. RhoA responded as predicted by manipulation of its regulators and that the biosensor(s) detected active RhoA; (ii) steady state levels can also be expressed as the ratio active/total pool; (iii) the model proposed does not consider/include the premise of how active pool versus total pool localized at junctions changes upon different stimuli; and (iv) the quantification correction in fig 2 (junctional over cytoplasmic levels of the biosensor) assesses enrichment of the biosensor at cell-cell contacts. Furthermore, the importance of controlling for specific activity can be found in the revised version, where the authors successfully demonstrated the amount of pY-cortactin corrected by levels of different cortactin constructs found at cell-cell contacts (Suppl Figure 1 f, h). Similar controls should be done for RhoA.

Response:

1) We welcome the opportunity to clarify this further. We certainly do not wish to argue that the junctional RhoA staining equates to active RhoA (i.e. that all the RhoA at the ZA is in the active form). For our studies we simply wished to establish that junctional RhoA signaling is compromised under a number of experimental conditions that are ultimately consistent with SRGAP1 acting to down-regulate RhoA at the junctions. Measurement of RhoA signaling is challenging, which is why we've elected to use a matrix of assays, from the AHPH biosensor to downstream measures of junctional contractility. In this analysis we have used changes in RhoA staining as a proxy index that the pathway is altered, not as a quantitative measure of the degree of change. We have endeavoured to modify our language to make this clearer.

2) The issue of specific activity (or, at the population level, the proportion of junctional RhoA which is active) is certainly an interesting problem. However, we were not sure how to effectively perform the experiment that the reviewer suggested. We have considered the following options.

A) AHPH: Although this sensor allows us to localize endogenous GTP-RhoA, it doesn't allow us to access the pool of RhoA that is not GTP-loaded and comparison of AHPH with e.g. RhoA staining is difficult to standardize reliably. Measuring the ratio of AHPH/total RhoA signal would not be a direct measure of the proportion of active RhoA as it would entail comparisons across different fluorophores and imaging methods (GFP-AHPH was imaged in PFA-fixed cells and RhoA was imaged in TCA-fixed cells). Moreover, any attempt to use the AH domain in a pull-down assay would eliminate the spatial information (i.e. the junctional pool) that is central to this study. (Moreover, the interaction between the anillin AH domain and GTP-RhoA is a very low-affinity interaction [~ 7.3 μ M and therefore potentially sensitive to detergents.]

B) We attempted to use antibodies that have been said to be specific for GTP-RhoA (NewStead Bioscience), which we might be able to normalize to total RhoA levels. However, in our hands we could not use it to identify any subcellular patterns of staining, despite using a range of fixation and preparation conditions.

C) The approach that we undertook was to quantitate the change in RhoA levels when its activity was blocked with C3-transferase. As seen in Supplementary Fig 3f,g, total protein levels (measured by quantitative IF) are reduced by $\sim 50\%$ upon C3-T treatment. This would suggest that

roughly 50% of the total RhoA pool may depend on RhoA being active, and, by implication, GTP-loaded. This gives a rough estimate of “specific activity”. Interestingly, we found that cortactin KD reduced AHPH levels by ~ 50% and total RhoA staining by ~ 25%. If we assume that half of the total RhoA pool is in the GTP-loaded state, this would equate to a reduction of the active pool by ~ 50%, comparable to the changes observed with AHPH. We are reluctant to apply this analysis further, though, as it assumes that the change in RhoA localization with C3T is a direct result of GTP-loaded status, whereas it could be more indirect. Again, as we only use this assay as a proxy index for changes in the RhoA signaling pathway, we feel that to take this further goes beyond a reasonable use of the assay.

3) It is possible that the presentation of our model has been a source of misunderstanding. It is certainly true that in the model we explicitly assume that *changes in RhoA activity can lead to changes in the amount of RhoA at the cortex*. However, the model is not designed to assess the relative proportion of total RhoA that is active or inactive. Instead, we designed the model to analyse the steady-state concentrations of **active** species at the junctional cortex (i.e. molecules that can engage downstream signaling and effectors). Therefore, the model only analyses changes in the amount of active species. We have revised the presentation of the model to make this more explicit.

We should emphasize that the question that we sought to address with the model was how SRGAP1 may affect the bistable behavior of the signaling network that controls the RhoA zone at the zonula adherens. We had earlier found that bistability is an important property that can influence the RhoA zone at the ZA (Priya et al., NCB 2015; PLoS Computational Biology, 2017), but what impact SRGAP1 might have on this was an open question. To explore this question in the model, the essential component was the net amount of active RhoA present at the junctions (i.e. whether steady-state levels of active molecules were altered).

Overall, we feel that the issue of how the precise proportion of active/total Rho may be altered at junctions, while important, is subsidiary to the principal message of our paper. This is to identify the recruitment of SRGAP1 as an antagonist of RhoA at the ZA, something that we have done by combining the use of GFP-AHPH as a location biosensor, with proxies including junctional ROCK1 and TCA-resistant RhoA staining, and downstream outcomes on myosin and junctional tension. We hope that the reviewer understands our thinking.

Other points:

1. The use of SRGAP1 mutant depleted of the GAP domain is not the appropriate mutant to demonstrate GAP-dependence. Technically, full-length SRGAP mutated at the key catalytic residue in the GAP domain is commonly used. The GAP-deleted mutant is considered a dominant-negative construct as shown for other GAPs, and most likely works by preventing the localization of the endogenous SRGAP to junctions. Experiments presented in the rebuttal letter showing the localization of different domains of SRGAP1 at junctions confirm this possibility. Thus the new experiments confirm the requirement of SRGAP and its correct positioning at contacts to modulate RhoA. The interpretation of the results should be modified to account for this. In addition, the term “GAP-deficient” means catalytically inactive (i.e. point mutation to abolish catalysis). For clarity, “GAP-deficient” used in the main text should be changed to “GAP-depleted” to reflect the dominant negative nature of the construct used.

Response: There seems to be a little confusion here, as the GAP-depleted mutant was expressed on an SRGAP1 KD background (where endogenous SRGAP1 was depleted by >80-90%), so it should not have been acting as a dominant-negative. Furthermore, we show under these conditions that the GAP-depleted mutant localizes to junctions (Fig S5h), so that aberrant localization cannot explain its effect. Overall, it seems to us reasonable to interpret its lack of rescue as reflecting at least loss of its GAP capacity, rather than a dominant-negative effect. Of course, we cannot exclude other functions served by this region of the molecule.

We are very happy to modify the text to “GAP-depleted” and have done so.

2. The new intensity quantification of proteins at junctions using the three uppermost confocal sections (rather than a single confocal section) is an improvement against variability across samples and treatments. The authors should clarify which experiments were re-quantified using this method or if the new analysis was applied across all results.

Response: Our quantifications from the raw images were always only performed on junctions that were in focus, as we described in Methods section. The representative images of Fig1c, Fig5a and Fig9a are the maximum projection of the three apical-most confocal sections. We have now specified these in the figure legends and Methods.

Reviewer #4:

The efforts by Liang and Yap to respond to reviews of their manuscript are greatly appreciated. Especially informative changes include coimmunoprecipitation experiments in Figure 3 showing a biochemical interaction between cortactin and SRGAP1 that is dependent on cortactin phosphorylation state. Figure 4 is also strengthened by rescuing RhoA inhibition in SRGAP1 knockdown cells with both WT and GAP-deficient SRGAP1.

However, without addressing relative expression levels of proteins in cortactin knockdown experiments as well as improving image quality in a select number of immunofluorescence experiments, this paper cannot be recommended for acceptance.

Major Points:

1) While the efforts of the authors to control for cortactin transgene localization to adherens junctions (Fig. S1d,e) are noted, immunoblots in Figures S1a and S1b still show that transgenes are expressed at much lower levels of endogenous protein. In Figure S1a transgenes are expressed at much lower levels than cortactin in the control cells, and at levels comparable to the remaining endogenous protein after knockdown. In Figure S1b the transgenes are expressed at lower levels than the remaining endogenous cortactin. Experiments performed with this degree of knockdown and reexpression cannot sufficiently isolate the effects of the transgene from that of endogenous cortactin. In fact, data in Figure S1d showing cortactin localization to junctions in these cells could represent a combination of endogenous and transgenic protein based on immunoblots.

Response: Here this may reflect a misunderstanding (which we did not adequately clarify in our writing). One reason why the transgene levels are lower than endogenous cortactin in control cells is because, although the majority of KD cells consistently express the transgenes, nonetheless it is somewhat heterogenous. It should also be noted that there is a large non-junctional pool. This is why we used quantitation of fluorescence intensity to focus on the junctional pool. Here, we would emphasize two things:

i) As shown in our quantitation, endogenous **junctional** cortactin measured by fluorescence is homogeneously reduced by ~ 90% by RNAi. Therefore, as the levels of cortactin in the transfected cells are similar, this implies that ~ 90% of the junctional cortactin is being contributed by the transgenes. We think that this is a reasonable indicator that they are substantively contributing to regulation at junctions.

ii) What is ultimately important for interpretation of the cortactin mutants is how they compare to the level of **WT transgene** that is expressed. As noted above, these are expressed to similar levels at junctions (measured with the fluorescent protein tag), so we feel that these are reasonable comparisons for our analysis. Indeed, in this revised version we now added statistical comparison of junctional expression of the WT transgene and cortactin mutant transgenes (Supplementary Figure 1e,f).

2) As mentioned by Reviewer 1, image quality needs to be improved, especially with regards to the pY421 antibody used for immunofluorescence. Images in Figures S1c, S1g, 6l, and S9b are examples of anti-pY421 blots that have high background or a speckled appearance that make it difficult to interpret quantification based on peak intensity in the adherens junction at points identified by the user. Interpretations of these images could be strengthened by overlaying another marker of the junction in a merged image, showing where line traces were performed, or comparing cytoplasmic to junction signal as the authors demonstrated in their rebuttal. Anti-Arp2/3 images in Figure S2C could benefit from the same methods.

Response:

We thank the reviewer for this suggestion. We have now added a merged-channel image of pY421 cortactin and E-cadherin in Fig. S1c. In Fig S1g (now S1h), as all three channels were already taken (by the pY421-cortactin, cortactin and tag imaging), we could not add an additional junctional marker (although we find that, like F-actin, cortactin itself is a clear junctional marker). Accordingly, we have included costaining of pY421 cortactin and E-cadherin in cortactin KD and reconstitutions cells (now Fig. S1g). These images show clear junctional pY421 cortactin staining that was reduced by cortactin KD or reconstitution with the 3YF mutant after KD. Furthermore, the junctional intensity values of pY421 cortactin were identified from intensity profiles for cortactin staining, as cortactin shows clear junctional staining. We compared the intensity values of pY421 cortactin to the same regions where cortactin showed its peak intensity values. We have now added this description in the Methods section.

We also added a channel of E-cadherin in the representative images of Fig. 6i and Fig. S9b (now Fig. S10b), and Arp3 and E-cad co-staining in Fig. S2c (now Fig. S3d), as the reviewer suggested.

Minor Points:

1. It is agreed that identifying the relevant cortactin phosphatase may be outside the scope of this work. However, authors should take care to not refer to “dephosphorylated cortactin” in their text unless supported by the experiment. Work done in Figure 3C using lambda phosphatase shows evidence of the effects of dephosphorylation on cortactin-SRGAP1 interaction. However, experiments performed using a 3YF mutant observe behavior of nonphosphorylatable cortactin, and do not directly speak to dephosphorylation events regulating Rho signaling.

Response: Agreed and where we refer to the 3YF mutant, we’ve now described it as “non-phosphorylated” cortactin, rather than “dephosphorylated” cortactin.

2. Efforts to perform experiments in Figures 3c-e are much appreciated and increase the rigor of this work significantly. A better representative immunoblot in Figure 3c could be chosen to reflect a decrease in pY421 upon lambda phosphatase treatment shown in the quantification in Figure 3d.

Response: We now included a better exposure of pYcortactin blots from the same experiment in Fig. 3c.

3. The added detail in the Methods section with regards to the interaction screen is very informative. However, if scores of interactions of phosphorylated cortactin with SRGAP1 as well as other top hits were included this would be both informative and impactful to the field. In the body of the text, mentioning the screen is mass spectrometry-based may also be helpful to readers.

Response: We have amended the description in Results to indicate that this was a “mass spectrometry-based interaction screen”. As we have not validated these hits extensively (our attention being caught by SRGAP1 as a potential RhoA antagonist), we would prefer to make these available to interested colleagues, on request.

4. Experimental details for Movies 1 and 2, including plating, labeling, and imaging conditions should be added to the Methods section.

Response: We now include culture conditions, experimental conditions, imaging conditions and labeling in Supplementary Methods section (“Live cell imaging”) and the movie legends of these movies.

REVIEWERS' COMMENTS:

Reviewer #3 (Remarks to the Author):

REvised manuscript is improved and addressed the previous concerns. The authors toned down the previous assertions about the majority of RhoA resident at junctions is activated(which was not substantiated by the data).

Regarding the relative amount of active RhoA at junctions, the authors are correct, the way the model was described is a source of misunderstanding (levels of RhoA = levels of activation). It is not well explained in the text as done in the rebuttal letter: the limitation that only active RhoA species are considered, rather than total amount of protein recruited/present at junctions and does not take into account in situ activation.

Would this assumption of "only evaluating changes in levels of active species" also apply to SRGAP1, an enzyme that requires itself activation and appropriate regulation in situ to inactivate RhoA? The conclusions of the model seem to consider SRGAP1 as a protein, just stating its presence or absence. i.e. page 12 "...junctional GTP-RhoA signalling zone is maintained despite the presence of SRGAP1" or "the threshold for SRGAP1 to enter its second regime may reflect when sufficient SRGAP1 has accumulated to overcome...".

This can be confusing and conceptually inaccurate. Prior to the description of the model, the authors should delineate precisely the assumptions on active species, and whether or not it applies also to SRGAP1. If the latter is considered in the model, the wording of the discussion and results should be modified to avoid confusion.

Reviewer #4 (Remarks to the Author):

The response by Liang and Yap to reviewer questions, including additional staining and explanations of experimental setup and image processing, is very thorough. The additional language in the text and methods sections improves the clarity of the work.

As Reviewer 3 mentions, the question of the ratio of endogenous to cortactin transgenes present at cellular junctions with respect to proportion of active vs. total RhoA at junctions is still a bit unclear.

The authors describe that the blot in Figure S1b, which shows higher levels of endogenous cortactin than transgenes in knockout cell lysates, is not indicative of levels at the junction, based on staining and analysis in Figures S1d-f. The authors' explanation that there is heterogeneity in expression of transgenes within the cellular population that results in the relatively low level of transgene expression seen by Western blot. This explanation is well appreciated and should be included in the figure legend or text.

However, the authors' conclusion that 90% of cortactin in junctions is transgenic and not endogenous, based on Figure S1e, would present three possible explanations:

- 1) The vast majority of knockout cells do not express the transgene. This could be addressed by including a larger field of view indicative of the relative expression of transgenes in cells to reconcile Figures S1b and S1e.
- 2) The non-junctional pool of endogenous cortactin is much higher than that of transgenic protein, which would be surprising given the lag of evidence that the tag alone is localizing to junctions.
- 3) Some combination of heterogeneity and localization differences between endogenous and rescue proteins.

The predominant concern with the assumption that 90% of junctional cortactin is transgenic is the amount of residual pY421 signal for the 3YF cells in Figure S1i. The signal for pY421 remains at 60% of phosphorylation relative to WT rescue, while the authors would expect a result closer to 10%. This could be due to much higher concentrations of endogenous cortactin at the junction than the authors anticipate, or nonspecificity of the antibody for pY421 (which was addressed in previous reviewer comments).

The authors should at a minimum consider directly addressing their findings in Figure S1i in the text and offering an explanation for higher than expected junction pY421 in 3YF cells.

Minor points:

1. What is the labeling method for cells in live cell imaging experiments, especially the blue colorization in wound healing assays?

2. Are the authors surprised by the presence of SRGAP1 in leading edge lamellipodia given the high degree of cortactin phosphorylation in those structures? This is a point that could be addressed in the discussion.

RESPONSE TO REVIEWERS

Reviewer #3 (Remarks to the Author):

Revised manuscript is improved and addressed the previous concerns. The authors toned down the previous assertions about the majority of RhoA resident at junctions is activated(which was not substantiated by the data).

Regarding the relative amount of active RhoA at junctions, the authors are correct, the way the model was described is a source of misunderstanding (levels of RhoA = levels of activation). It is not well explained in the text as done in the rebuttal letter: the limitation that only active RhoA species are considered, rather than total amount of protein recruited/present at junctions and does not take into account in situ activation.

Would this assumption of “only evaluating changes in levels of active species” also apply to SRGAP1, an enzyme that requires itself activation and appropriate regulation in situ to inactivate RhoA? The conclusions of the model seem to consider SRGAP1 as a protein, just stating its presence or absence. i.e. page 12 “...junctional GTP-RhoA signalling zone is maintained despite the presence of SRGAP1” or “the threshold for SRGAP1 to enter its second regime may reflect when sufficient SRGAP1 has accumulated to overcome....”.

This can be confusing and conceptually inaccurate. Prior to the description of the model, the authors should delineate precisely the assumptions on active species, and whether or not it applies also to SRGAP1. If the latter is considered in the model, the wording of the discussion and results should be modified to avoid confusion.

Response: We have now added the following sentence to introduce the model in the Results section:

“As discussed in the Supplementary Information, this model assumes that proteins localized at the junctional cortex are active, including SRGAP1.” (p 9, bottom para)

The model and its assumptions are described in greater depth in the Supplementary Information. For conciseness, where we later discuss SRGAP1, we have left it as assumed that we are dealing with the signaling-active species, since in the paper we do not deal with mechanisms that regulate the intrinsic activity of SRGAP1.

Reviewer #4 (Remarks to the Author):

The response by Liang and Yap to reviewer questions, including additional staining and explanations of experimental setup and image processing, is very thorough. The additional language in the text and methods sections improves the clarity of the work.

As Reviewer 3 mentions, the question of the ratio of endogenous to cortactin transgenes present at cellular junctions with respect to proportion of active vs. total RhoA at junctions is still a bit unclear.

The authors describe that the blot in Figure S1b, which shows higher levels of endogenous cortactin than transgenes in knockout cell lysates, is not indicative of levels at the junction, based on staining and analysis in Figures S1d-f. The authors' explanation that there is heterogeneity in expression of transgenes within the cellular population that results in the relatively low level of transgene expression seen by Western blot. This explanation is well appreciated and should be included in the figure legend or text.

Response: We have added a note to the caption for Fig S1a:

“(a) Immunoblots of non-muscle myosin IIA (NMIIA), E-cadherin (E-cad), cortactin in cortactin KD and mutants reconstituted Caco-2 cells. GAPDH was used as a loading control. Molecular marker unit, kDa. Overall expression levels of cortactin transgenes are lower than endogenous cortactin as not all KD cells expressed the transgenes.”

However, the authors' conclusion that 90% of cortactin in junctions is transgenic and not endogenous, based on Figure S1e, would present three possible explanations:

- 1) The vast majority of knockout cells do not express the transgene. This could be addressed by including a larger field of view indicative of the relative expression of transgenes in cells to reconcile Figures S1b and S1e.
- 2) The non-junctional pool of endogenous cortactin is much higher than that of transgenic protein, which would be surprising given the lag of evidence that the tag alone is localizing to junctions.
- 3) Some combination of heterogeneity and localization differences between endogenous and rescue proteins.

The predominant concern with the assumption that 90% of junctional cortactin is transgenic is the amount of residual pY421 signal for the 3YF cells in Figure S1i. The signal for pY421 remains at 60% of phosphorylation relative to WT rescue, while the authors would expect a result closer to 10%. This could be due to much higher concentrations of endogenous cortactin at the junction than the authors anticipate, or nonspecificity of the antibody for pY421 (which was addressed in previous reviewer comments).

The authors should at a minimum consider directly addressing their findings in Figure S1i in the text and offering an explanation for higher than expected junction pY421 in 3YF cells.

Response: The reviewer makes a good point. The discrepancy between Fig S1b and S1i is likely to reflect background staining by the antibody. Accordingly, we have modified the text (bottom, p4) to note this:

“However, although the transgenes were expressed at junctions to similar levels, pY421-staining was reduced in cells expressing 3YF cortactin compared with either WT or 3YD cortactin (some residual staining may reflect background staining,

Supplementary Fig. 1g-j).”

Minor points:

1. What is the labeling method for cells in live cell imaging experiments, especially the blue colorization in wound healing assays?

Response: Nuclei were labeled with NucBlue. We neglected to mention this in the Methods, which have now been corrected accordingly.

2. Are the authors surprised by the presence of SRGAP1 in leading edge lamellipodia given the high degree of cortactin phosphorylation in those structures? This is a point that could be addressed in the discussion.

Response: It is possible that SRGAP1 is being recruited to leading edges in response to a different mechanism. For example, Yamazaki et al., (MBoC, 2013) found a dominant role for Rac in recruiting SRGAP1 to the leading margins of fibroblasts. This would be consistent with other evidence (noted in the Discussion) that multiple mechanisms may recruit SRGAP1 to the cortex in different circumstances. Accordingly, we have added a sentence in the Discussion (top, p14):

“This may also pertain in our experimental model, as SRGAP1 was also seen in lamellipodia of migrating cells.”